# Data-Free Diversity-Based Ensemble Selection for One-Shot Federated Learning

**Naibo Wang[1], Wenjie Feng[1, †], Yuchen Deng[2, †], Moming Duan[1], Fusheng Liu[1], See-Kiong Ng[1]**

*{naibowang, fusheng}@u.nus.edu, {wenjie.feng, moming, seekiong}@nus.edu.sg, dengyuchen.cc@gmail.com*
[1] *Institute of Data Science, National University of Singapore*
[2] *School of Mathematics and Statistics, Changchun University of Technology*
†: Corresponding author.

**Reviewed on OpenReview:** `https://openreview.net/forum?id=ORMlg4g3mG`

## Abstract

The emerging availability of various machine learning models creates a great demand to harness the collective intelligence of many independently well-trained models to improve overall performance. Considering the privacy concern and non-negligible communication costs, one-shot federated learning and ensemble learning in a data-free manner attract significant attention. However, conventional ensemble selection approaches are neither training efficient nor applicable to federated learning due to the risk of privacy leakage from local clients; meanwhile, the *many could be better than all* principle under data-free constraints makes it even more challenging. Therefore, it becomes crucial to design an effective ensemble selection strategy to find a good subset of the base models as the ensemble team for the federated learning scenario. In this paper, we propose a novel *data-free diversity-based* framework, `DeDES`, to address the ensemble selection problem with diversity consideration for models under the one-shot federated learning setting. Experimental results show that our method can achieve both better performance and higher efficiency over 7 datasets, 5 different model structures, and both homogeneous and heterogeneous model groups under four different data-partition strategies.

## 1 Introduction

In the era of large models with great appetites for large-scale data for various machine learning tasks, to handle the data island while addressing the increasing demands on data privacy concern for information sharing, federated learning (FL) (Li et al., 2020) has become the mainstay for enabling collaborative machine learning on decentralized devices/parties without accessing private data.

Traditional federated learning often requires a multi-round training process, and $O(mn)$ gradients or models will be acquired for the scenario with $m$ clients and $n$ update rounds, resulting in great potential risk for the leakage of local data and violation of the privacy-preserving principle (Zhu et al., 2019; Geiping et al., 2020). One-shot federated learning (Guha et al., 2018; Su et al., 2023) is proposed to alleviate the above issues, which only requires the clients to send their well-trained models to the server once, while the communication costs are significantly reduced. However, it is foreseeable that the performance of the resulting models in one-shot FL will often be inferior to that in conventional FL (Yurochkin et al., 2019; Lam et al., 2021). As a consequence, one-shot FL models cannot be competent in some critical and highly demanding fields, such as medical diagnosis and financial regulation.

Other model-centric research lines include model fusion (Kasturi et al., 2020), knowledge distillation (Zhu et al., 2021), and ensemble learning (Zhou, 2021; Sagi & Rokach, 2018). Of these, ensemble learning is straightforward and cost-effective harnessing the power of collective models to boost task performance in a data-free manner. For instance, majority voting, the commonly used classical method, produces the final prediction results depending on the voting of multiple models. However, given the principle that *many*

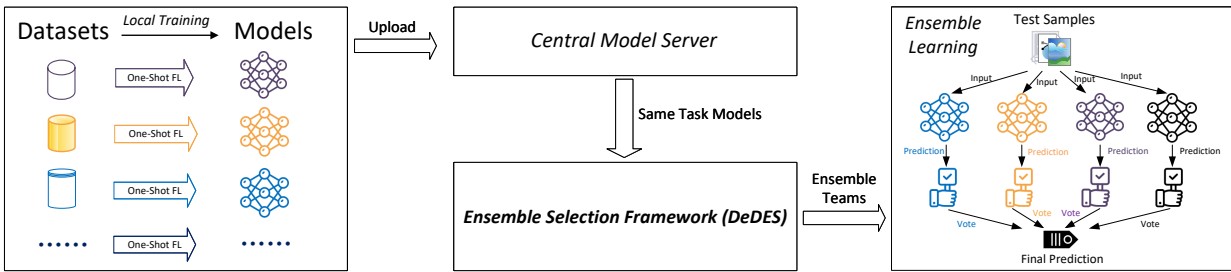

Figure 1: Overview of ensemble learning and ensemble selection process under one-shot federated learning.

*could be better than all* (Zhou et al., 2002), it may not always be the most effective strategy for the central server in FL to select all available client models, especially for neural network models. Meanwhile, it is time-consuming and inefficient to test each incoming sample on all models when the ensemble team contains a large number of models. Hence, to ensure the performance of the aggregated model in one-shot FL, it is crucial to maximize population divergence within the constraints of population size (Wu et al., 2021; Wu & Liu, 2021). Unfortunately, the existing diversity computation methods proposed so far typically require accessing the local datasets of the clients, which is unsuitable for FL scenarios. Therefore, the designing of an effective and computation-efficient ensemble selection strategy as well as new computation methods for model diversity for one-shot FL are challenging problems.

In this work, we focus on the ensemble selection problem under the one-shot FL setting and aim to find a near-optimal subset of models to the ensemble for the central model server in a data-free manner. As the basic workflow shows in Figure 1, each client trains its local model with their own private dataset and then uploads the well-trained models to the central model server. To perform the ensemble learning, the server will select the best ensemble team from all models. Note that during the whole process, the server has no access to the local datasets of clients, which actually follows the *data-free* manner.

We propose a novel framework, Data-free Diversity-based Ensemble Selection (`DeDES`), to select the best ensemble team from the model set trained in the one-shot FL fashion, which is a clustering-based ensemble selection method to ensure the diversity without access to any public or client-side dataset. Extensive experiments show that DeDES achieves the best performance for different settings, and is robust and effective. Also, it can adapt to various datasets, data partitions (including the non-iid case), and model structures (both homogeneous and heterogeneous models). To the best of our knowledge, this is the first effort to systematically deal with the ensemble selection issues for one-shot FL.

Our main contributions can be summarized as follows,

- We provide the formulation of the ensemble selection problem for one-shot Federated Learning to facilitate clearer comprehension of the topic.
- We propose a novel one-shot Federated Learning framework `DeDES` equipped with the data-free ensemble selection strategy, which can evaluate model diversity and conduct ensemble selection in a data-free manner for privacy-preserving consideration. Our method is model-agnostic and can be applied to both homogeneous and heterogeneous model groups.
- We conduct comprehensive experiments to verify the effectiveness and efficiency of our `DeDES`, which can also adapt to both homogeneous and heterogeneous settings. In comparison to the state-of-the-art data-free one-shot federated learning framework, our proposed model ensemble methodology demonstrates a substantial enhancement in performance efficacy.

## 2 Related Work

### 2.1 Federated Learning and One-Shot Federated Learning

Various federated learning systems (Bonawitz et al., 2019; Duan et al., 2020) have been proposed to assist multiple parties in cooperatively working with others without disclosing their data. E.g., Wu et al. (Wu

et al., 2023) addressed the client drift problem on personalized FL by using a GMM to fit the joint data distributions across FL devices. In particular, one-shot federated learning (Guha et al., 2018; Su et al., 2023; Feng et al., 2023), including FedKT (Li et al., 2021), Fusion Learning (Kasturi et al., 2020) and so on, aims to facilitate collaboration through a single round of communication between the server and clients. Su et al. (Su et al., 2023) proposed MA-Echo that employs layer-wise projection matrices to maintain the original loss of local models after aggregation of parameters, which requires all model structures to be the same; FedDISC (Yang et al., 2023) explored one-shot semi-supervised federated learning with a pre-trained diffusion model. Furthermore, with the popularity of pre-trained models, there are emerging interests in collaborative model-centric machine learning (Bommasani et al., 2021) as a platform for users to exchange their trained models with others, and to harness collective intelligence for the targeted machine learning task by combining models. Diao et al. (Diao et al., 2022) and Joshi et al. (Jhunjhunwala et al., 2023) separately explored one-shot federated learning through the open-set problem and Fisher information. However, neither addressed the model selection problem that is critical for large-scale FL systems, which makes our approach distinct and orthogonal to their methods.

## 2.2 Ensemble Learning and Ensemble Selection

Ensemble learning, which seeks to combine multiple weak base models into a strong model, has been a popular research topic for decades that can be applied to federated learning scenarios. Classical ensemble learning approaches include Voting (Zhou, 2021), Bagging (Breiman, 1996; Sagi & Rokach, 2018), Boosting (Schapire, 1990; 2013), and Stacking (Wolpert, 1992; Wang et al., 2019).

As an important step in ensemble learning, *ensemble selection* needs to select an ensemble team from the whole model set for each test sample; there exist three types of ensemble selection approaches, i.e., search-based (Caruana et al., 2004), rank-based (Ma et al., 2015), and cluster-based (Maskouni & Zhou, 2018). Notably, the cluster-based method stands out for its superior efficacy, which primarily stems from its reliance on model diversity. Luis A et al. (Ortega et al., 2022) give a theoretical analysis that the expected loss of an ensemble *decreases* as the *diversity* measure among model ensembles *increases*. Classic model diversity computation methods include Binary Disagreement (Kuncheva & Whitaker, 2003), Cohen's Kappa (McHugh, 2012), Q Statistics (Zhang & Cao, 2014), Generalized Diversity (Partridge & Krzanowski, 1997), and Kohavi-Wilpert Variance (Kuncheva & Whitaker, 2003). All of them require accessing the local dataset and cannot be directly used for federated learning.

## 2.3 Knowledge Distillation in Federated Learning

Another field of interest for federated learning is knowledge distillation (Lin et al., 2020; Gong et al., 2021). FedGen (Zhu et al., 2021) is a data-free method for heterogeneous federated learning that necessitates multiple rounds of communication between the server and clients. DENSE (Zhang et al., 2022) is a data-free one-shot method to train a global model, and it involves grouping all clients as an ensemble, which may lead to decreased performance and efficiency. Thus, our ensemble selection approach could potentially assist in improving their performance and training efficiency. FedCAVE-KD (Heinbaugh et al., 2023) is an instance of knowledge distillation generation. In this method, locally trained CVAEs and local label distributions are uploaded to a server for data-free knowledge distillation, which ensures privacy and simultaneously improves the generalization of the global model. However, they only send the decoder instead of CNN to the server, so their models cannot be directly used to conduct ensemble, not to say ensemble selection. To conclude, none of the existing methods tackle the ensemble selection problem for one-shot federated learning.

## 2.4 Client Selection in Federated Learning

Several research works focus on the issue of client selection in federated learning (AbdulRahman et al., 2020), Nishio et al. (Nishio & Yonetani, 2019) enables the server to combine as many client updates as feasible within a specified time-frame; Cho et al. (Cho et al., 2022) presented a computation-efficient client selection framework to alleviate the bias of model aggregation; Huang et al. (Huang et al., 2020) proposed an efficiency-boosting client selection scheme to guarantee the fairness of the training process. However, those methods are independent of our ensemble selection topic and cannot be applied to our problem.

## 3 Problem Definition

We assume that there are $m$ different parties (aka FL clients) and a central model server in our one-shot federated learning system, where the parties want to collaborate together on a given machine learning task, e.g., classification or regression, and the server builds the model ensemble team for one-shot federated learning. Let $\mathcal{M} = \{M_1, \ldots, M_m\}$ be the model set, in which $M_i$ is well-trained on the $i$-th client over its own private dataset $D_i = \{(x_k, y_k)\}_{k=1}^{n_i}$ with size $n_i$, where each data vector is sampled from an unknown distribution $\mathcal{D}$. Note that $\mathcal{M}$ will be uploaded by each party to the central model server. Hence, the ensemble selection problem can be formulated as:

**Problem 1** *Given the model set $\mathcal{M}$ and the constant $K \ll m$, find the optimal subset $\mathcal{M}_K^* \subset \mathcal{M}$ such that*

$$\mathcal{M}_K^* = \underset{\mathcal{M}_K \subseteq \mathcal{M}, |\mathcal{M}_K| = K}{\arg\min} \mathbb{E}_{(x,y) \sim \mathcal{D}} \ell(f_{\mathcal{M}_K}(x), y), \tag{1}$$

*where $f_{\mathcal{M}_K}(\cdot)$ is the prediction function based on $\mathcal{M}_K$, $\ell$ is the loss function, e.g., MSE loss function for regression tasks and Cross-Entropy loss function for classification tasks.*

Under the ensemble learning setting, $f_{\mathcal{M}_K}$ is the aggregation function to combine the prediction of $M_i \in \mathcal{M}_K$ for the final prediction $\hat{y} = f_{\mathcal{M}_K}(x)$; it can be weighted average for regression, or weighted voting-based (e.g., majority or plurality voting) for classification. Under the model fusion setting, $f_{\mathcal{M}_K}$ is the prediction of the fused model based on all models in $\mathcal{M}_K$.

In the following sections, we mainly focus on the classification task and adopt the weighted voting strategy based on the size of local clients' datasets for ensemble learning. For a $C$-class classification (i.e., the label set is $\{1, \ldots, C\}$) task, with $\mathbb{I}(\cdot)$ as the indicator function, the prediction $\hat{y}$ of the input $x$ is given by

$$\hat{y} := \underset{c \in \{1, \ldots, C\}}{\arg\max} \sum_{j=1}^{K} \frac{n_i}{\sum_{k=1}^{K} n_k} \mathbb{I}\left(M_j(x) = c\right), \tag{2}$$

## 4 Proposed Framework: `DeDES`

We present the proposed ensemble selection framework, `DeDES`, to solve the Problem 1 without accessing any private dataset from local clients. Algorithm 1 summarizes the structure of `DeDES`, which can be adapted to heterogeneous models.

Considering the performance and efficiency of $\mathcal{M}_K^*$, it is necessary to choose a small $K$ while keeping the diversity and high-quality among selected elements/models. `DeDES` achieves such goal via the following key components: *model filtering*, *model representation*, *model clustering*, and *representative model selection*.

### 4.1 Key Components in `DeDES`

***Model filtering:*** Coming from multiple parties in FL, the test performance of local models can vary significantly and is out-of-control to the central server. Possible inferior models may result from different reasons, including low-quality training data (e.g., being unreliable, contaminated, or noisy), inefficient training (e.g., trained with improper hyperparameters), etc. Therefore, it is necessary to filter out such outlier models to eliminate the effect of the noise and help to select high-quality models efficiently. In Alg. 1, we use the `OutlierFilter` to obtain the outlier models $\mathcal{O}$ based on the model scores $\mathcal{S}$ provided from each party. Note that $\mathcal{S}$ could be the local validation accuracy or prediction confidence. `OutlierFilter` could be any score-based unsupervised outlier detection methods (Zhao et al., 2019). In Alg. 2, we implement a variation of the box-plot method to identify the outliers when their scores are lower than the threshold $\delta$.

***Model representation:*** Given the model structure and its parameters, generating effective and suitable representations for the models is crucial to measure their properties, such as similarity and diversity. In Alg. 1, we obtain the representations via the function `Representation` in Line 4. Intuitively, we can

---

**Algorithm 1:** `DeDES` framework

---

**Input:** model set $\mathcal{M}$, training-set sizes $\mathcal{N} = \{n_i\}_{i=1}^m$, truncated threshold pair $(p_{low}, p_{high})$, interval scale $s$, model scores $\mathcal{S} = \{s_i\}_{i=1}^m$, target model-set size $K$, and representative selection threshold $\tau$.

**Output:** Optimal model subset (ensemble team) $\mathcal{M}_K^*$.

**1** $\mathcal{M}_K^* \leftarrow \emptyset$

    ▷ *1. Model filtering: select high-quality candidates by filtering out outliers.*

**2** $\mathcal{O} \leftarrow \text{OutlierFilter}(\mathcal{M}, \mathcal{S}, (p_{low}, p_{high}), s)$                    ▷ *$\mathcal{O}$: outlier models set*

**3** $\mathcal{N}' = \{n_i \mid M_i \notin \mathcal{O}, \forall M_i \in \mathcal{M}\}$;   $\mathcal{S}' = \{s_i \mid M_i \notin \mathcal{O}, \forall M_i \in \mathcal{M}\}$;   $\mathcal{M}' = \mathcal{M} \setminus \mathcal{O}$;

    ▷ *2. Model representation: get model's feature representation.*

**4** $\mathcal{R}_{\mathcal{M}'} = \text{Representation}(\mathcal{M}')$

    ▷ *3. Model clustering: get K-size model clusters for diversity selection.*

**5** $\mathcal{C}_{\mathcal{M}'} = \text{Clustering}(\mathcal{R}_{\mathcal{M}'}, K)$

    ▷ *4. Representative model selection: choose the 'best' model in each cluster.*

**6** **for** $\mathcal{C} \in \mathcal{C}_{\mathcal{M}'}$ **do**

**7**      $\mathcal{N}_{\mathcal{C}} = \{n_i \mid \forall M_i \in \mathcal{C} \cap \mathcal{M}'\}$

**8**      $n_{\max}^{\mathcal{C}} \leftarrow \max(\mathcal{N}_{\mathcal{C}}); \; n_{\text{med}}^{\mathcal{C}} \leftarrow \text{median}(\mathcal{N}_{\mathcal{C}})$

         ▷ *$\tau$: user predefined threshold, e.g., $\tau = 0.3$.*

**9**      **if** $n_{\text{med}}^{\mathcal{C}} / n_{\max}^{\mathcal{C}} < \tau$ **then**

**10**         $k = \arg\max_j \{n_j \mid M_j \in \mathcal{C} \cap \mathcal{M}'\}$

**11**      **else**

**12**         $k = \arg\max_j \{s_j \mid M_j \in \mathcal{C} \cap \mathcal{M}'\}$

**13**      $\mathcal{M}_K^* \leftarrow \mathcal{M}_K^* \cup \{\mathcal{C}_k\}$                 ▷ *$\mathcal{C}_k$: the k-th element of the cluster $\mathcal{C}$*

**14** **return** $\mathcal{M}_K^*$

---

**Algorithm 2:** `OutlierFilter` algorithm for the model filtering

---

**Input:** model set $\mathcal{M}$, truncated threshold pair $(p_{low}, p_{high})$, interval scale $s$, model scores $\mathcal{S} = \{s_i\}_{i=1}^m$.

**Output:** Outlier model set $\mathcal{O}$.

**1** $\hat{\mathcal{S}} \leftarrow Sort(\mathcal{S})$                               ▷ *In a non-decreasing order*

**2** $\delta \leftarrow \min(\hat{\mathcal{S}}) + s * (\hat{\mathcal{S}}_{m \times p_{high}} - \hat{\mathcal{S}}_{m \times p_{low}})$

**3** $\mathcal{O} \leftarrow \emptyset$

**4** **for** $i = 1$ to $m$ **do**

**5**      **if** $s_i < \delta$ **then**

**6**         $\mathcal{O} \leftarrow \mathcal{O} \cup \{M_i\}$

**7** **return** $\mathcal{O}$

---

represent a model by all or part of its model parameters. E.g., we can choose the parameters of the last layer of the model, which contain individualized and sufficient information about the model behavior (especially for the classifier) and data manifold/space for local training. Meanwhile, to distill compact information and suppress noise for the representation, especially for big models such as Resnet-101 (Wu et al., 2019), dimension reduction (DR) is also applied to the representations; many unsupervised approaches can be adopted here, including PCA, Kernel-PCA, etc. The target dimension for DR is set to be $|\mathcal{M}'|$ by default.

***Model clustering:*** To guarantee the diversity in $\mathcal{M}_K^*$ as we mentioned before, we can utilize the clustering method to identify the similarity of different models, where models with similar properties are grouped into the same cluster and different clusters are as different as possible. We can use the traditional clustering approach here, such as KMeans, Hierarchical Clustering, and Spectral Clustering, etc., and set the target number of clusters as $K$. This process is denoted by `Clustering` in Alg. 1, which leads to $\mathcal{C}_{\mathcal{M}'}$ as the resultant clusters.

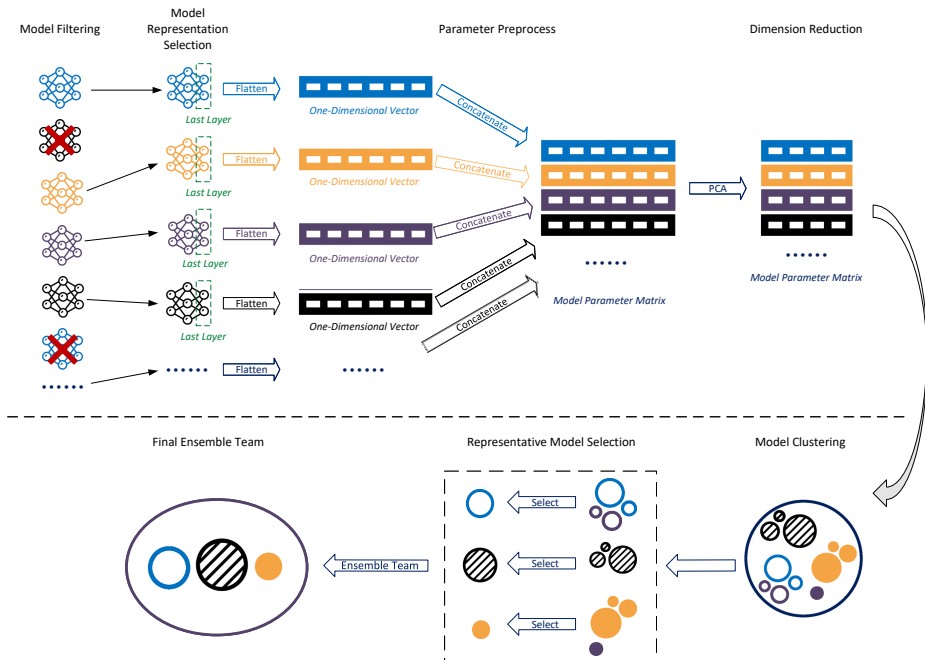

Figure 2: The flow chart of the execution processes of `DeDES` framework. All model architectures are identical and the last layer is used to represent the model. In this flowchart, PCA is used as the dimension reduction method. The circles of the same style with the same color and texture represent models with actual high similarity. The model parameter matrix in the figure is the model representation $\mathcal{R}_{\mathcal{M}'}$ mentioned in Alg. 1.

***Representative model selection:*** To choose exactly $K$ models with high performance, we elaborately select the representative element in each cluster while keeping the diversity. Among the models in each cluster, we can intuitively select the model with either the highest model score $s_i \in \mathcal{S}$ (provided by the individual party) or the largest training dataset (leading to a better-trained model). As Line 6-13 in Alg. 1 shows, we design a heuristic selection strategy to make full use of these two ways, which can choose a better one than any of the fixed ways as the experiment results proves. That is, if the amount of training data for the models inside the cluster is balanced, the model with the highest score is chosen, otherwise, the one with the largest training data is chosen.

Figure 2 presents a flow chart that exemplifies the implementation of `DeDES` using the parameters of the last layer as the model representation and PCA as the method of dimension reduction when all model structures are identical. The clustering method is dependent on the employed data partition strategies.

***Inference:*** After obtaining the optimal $\mathcal{M}_K^*$ with Algorithm 1, we will conduct ensemble learning with the weighted voting as Eq. 2. Note that in the whole process of `DeDES`, we successfully select the ensemble team $\mathcal{M}_K^*$ based on the *model diversity* without accessing any of *local private data* of these parties.

## 4.2 Adaptation to Heterogeneous models

When the structures of the client models are different, i.e., heterogeneous models, our method transforms them into a unified model parameter matrix $\mathcal{R}_{\mathcal{M}'}$.

Assuming that there are $s$ types of model structures and each contains any number of models, the model set $\mathcal{M}'$ for all those models is,

$$\mathcal{M}' = \{M_1, M_2, \ldots, M_{m'}\} \xrightarrow{\text{group by } s} \bigcup_{i=1}^{s} \mathcal{M}'_i \text{ with } \mathcal{M}'_i = \{M_{i,1}, \ldots, M_{1,m'_i}\},$$

where $m'_i$ denotes the number of models in $\mathcal{M}'_i$ that share the $i$-th model structure; and the total number of models in $\mathcal{M}'$ is $m' = \sum_{i=1}^{s} m'_i$.

We extract part of the model parameters and flatten them to be a one-dimensional vector as the representation of the model. For the model $M_i \in \mathcal{M}'$, let $\boldsymbol{\theta}_i$ denote the flattened selected parameters (e.g., the parameters of the last layer) of $M_i$. Let $V_i = \{\boldsymbol{\theta}_{i,1}, \ldots, \boldsymbol{\theta}_{i,m'_i}\}$ be the vector set for the model subset $\mathcal{M}'_i$. Thus, the vector set $V$ for all models $\mathcal{M}'$ will be $V = \bigcup_{i=1}^{s} V_i = \{\boldsymbol{\theta}_1, \boldsymbol{\theta}_2, \boldsymbol{\theta}_3, \ldots, \boldsymbol{\theta}_{m'}\}$.

Since the model structures vary for different $\mathcal{M}_i$, the sizes of the vectors in $V$ are also different, which makes it hard to directly apply the clustering methods. To solve such an issue, we reduce the dimension of all vectors to a unified dimension $d$. Given a $DR$ algorithm, we transform the size of all vectors $\boldsymbol{\theta}_i$ of $V$ into $d = \min_{\boldsymbol{\theta} \in V} |\boldsymbol{\theta}|$ by applying $DR$ to each sub-vector set $V_i$, separately. The *model parameter matrix* $\mathcal{R}_{\mathcal{M}'}$ is obtained by combining all these matrices together,

$$\mathcal{R}_{\mathcal{M}'} = [DR(V_1, d), DR(V_2, d), \cdots, DR(V_s, d)]^T. \tag{3}$$

$\mathcal{R}_{\mathcal{M}'} \in \mathbb{R}^{m' \times d}$ with each sub-matrix $DR(V_i, d) \in \mathbb{R}^{m'_i \times d}$. We can then perform further clustering on $\mathcal{R}_{\mathcal{M}'}$, since all heterogeneous models are transformed to the same dimensions.

### 4.3 Complexity analysis and Resource Requirements

**Lemma 1** *Given $m$ client models, $K$ target models, and $d$ as the size of the selected parameters of a single model, the overall time complexity of `DeDES` is $O(m \log m + md + m) + O(clustering)$, where $O(clustering)$ is the time complexity of the applied model clustering method.*

**Proof 1** *The time complexity of each component in `DeDES` is given as follows:*

*For the model filtering, it takes $O(m \log m)$ time due to sorting of model scores, which is the most time-consuming step; the model representation takes $O(md)$ time, and it linearly depends on the size of the selected parameters in a model. The complexity of the model clustering depends on the specific algorithm being employed, that is, it will be $O(mKd)$ if KMeans is used, and be $O(m^3)$ if adopting spectral clustering. The representative model selection step takes $O(m)$ time for it linear filtering.*

*Hence, when employing the KMeans in `DeDES`, the total time complexity is $O(m \log m + md + mKd + m)$.*

**Communication Cost.** Our approach adheres to the one-shot federated learning scheme, which inherently requires every client to communicate with the server **only once** - when they transmit their respective models.

## 5 Experiments

### 5.1 Experiment Setup

To simulate the real scenarios in FL (Li et al., 2022) we designed four types of dataset-partition strategies to evaluate `DeDES`, which lead to different local data distribution to train diverse client models $M_i$.

- Homogeneous (*homo*): the amount of samples and the data distribution keep the same for all parties.

- IID but different quantity (*iid-dq*): the training data of each party follows the same distribution, but the amount of data is different.

- Skewed data distribution (*noniid-lds*): the training data of each party follows different distributions, especially for the label distribution.

- Non-iid with $k$ ($< C$) classes (*noniid-lk*): the training data of each party only contains $k$ of $C$ classes, which is an extreme non-iid setting.

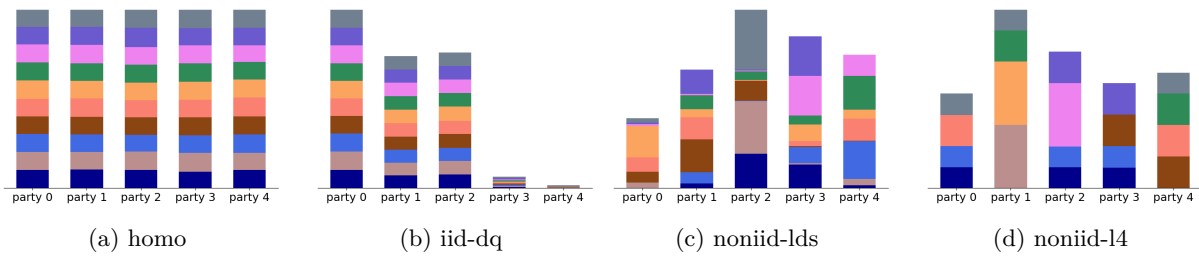

Figure 3: Example distribution of four data partition strategies for *CIFAR-10* with party number $m = 5$. Each color denotes one class, and the height of the bar represents the number of samples of that class.

We used 7 image datasets and 5 types of neural network models in our experiments, details can be found in the Appendix. We partition all datasets into different groups based on the above strategies and train the model for each client. Figure 3 shows an example of the data distribution under the different partition strategies for CIFAR-10 with 5 parties.

The detailed runtime setups and configuration of `DeDES` are elaborated in the Appendix, including the learning rate, model representation strategy, clustering method for different data partitions, etc.

## 5.2 Baselines

For the model ensemble learning under our problem setting, we follow the designs in the first one-shot FL framework (Guha et al., 2018) and summarize the well-known selection approaches as follows:

- *Cross-validation selection (CV)*: select $\mathcal{M}_K^*$ using local validation accuracy;
- *Data selection (DS)*: $\mathcal{M}_K^* = \{M_i \,|\, i \in \mathtt{top}(\{n_1, \cdots, n_m\}, K)\}$, i.e., the models trained with the top $K$ size training dataset, which are selected by $\mathtt{top}$;
- *Random selection (RS)*: $\mathcal{M}_K^*$ consists of $K$ models random selected from $\mathcal{M}$.

Besides, we construct well-known baselines in terms of model fusion, for comparison with the traditional FL methods. They derive *one* single model $M^*$ leading to the highest efficiency for inference as follows

- *One-Shot Federated averaging (FedAvg)*: $M^* = \sum_{i=1}^m \frac{n_i}{\sum_{j=1}^m n_j} M_i$;
- *One-Shot Mean averaging (MeanAvg)*: $M^* = \frac{1}{m} \sum_{i=1}^m M_i$.

Also, we include the following results as the ground-truths for comparison,

- *All selection (AS)*: select $\mathcal{M}$ as the target model set by ignoring $K$.
- *Label distribution selection (LDS)*: utilizing the label distribution instead of model representation as the input of our method [1];
- *Oracle*: using the aggregated dataset $D = \bigcup_{i=1}^m D_i$ to train a model $M_{oracle}$, which is unrealistic for the real FL scenario.

## 5.3 Performance Analysis

**The effectiveness of ensemble learning.** Figure 4 shows the comparison result for 4 types of data partition settings, where *TOP 1* and *TOP 2* mean a single model that gets the best and second test accuracy on the whole test dataset $D^{test} = \bigcup_{i=1}^m D_i^{test}$, where $D_i^{test}$ is the test set for $i$-th party/client. As we can see, the performance of the ensemble methods such as *AS* and `DeDES` are always better than single models, which validates the effectiveness of ensemble learning under the one-shot federated learning setting.

**Comparison of `DeDES` with other methods.** For $m$ models, the number of possible ensemble teams is $2^m$, which increases exponentially with $m$. Since testing all teams to get the optimal one is impractical unless $m$ is very small, we compare `DeDES` with other existing methods to demonstrate its superiority with the

---

[1]Note that the label distribution is unavailable in the real federated learning scenarios.

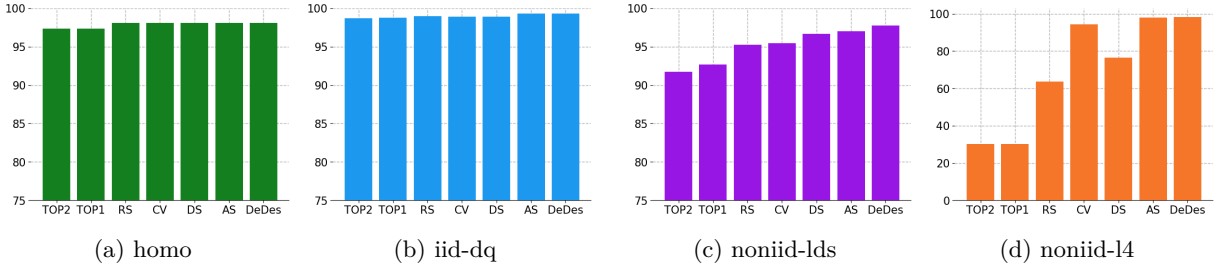

| (a) homo | (b) iid-dq | (c) noniid-lds | (d) noniid-l4 |

Figure 4: Ensemble learning (weighted voting strategy) performance comparison on *EMNIST Digits* dataset for $m = 200$, $K = 80$. The x-axis shows different ensemble selection method names, and the y-axis is the test accuracy (%) on the whole test dataset $D^{test} = \bigcup_{i=1}^{m} D_i^{test}$, where $D_i^{test}$ is the test set for $i$-th client.

Table 1: Test accuracy comparison for different datasets on various data partitions and model structures. The best and next-best are **bolded** and underlined, respectively. If our `DeDES` method is better than the ground-truth methods such as *AS* and *LD*, the value of the corresponding method are marked in skyblue.

| Dataset | Partition | $m$ | $K$ | DeDES | CV | DS | RS | FedAvg | MeanAvg | AS | LD | Oracle |
|---|---|---|---|---|---|---|---|---|---|---|---|---|
| EMNIST Digits (100% VGG-5 Spinal FC) | homo | 400 | 150 | 98.03 | **98.10** | 98.08 | 98.07 | 10.28 | 10.26 | 98.10 | 98.10 | 99.74 |
| | iid-dq | 400 | 150 | **99.27** | 98.93 | 98.88 | 98.72 | 10.51 | 10.48 | 98.75 | 99.27 | 99.71 |
| | noniid-lds | 400 | 150 | **97.67** | 95.47 | 91.70 | 96.67 | 10.01 | 9.89 | 96.99 | 92.86 | 99.72 |
| | noniid-l3 | 400 | 150 | **98.21** | 97.87 | 63.59 | 94.35 | 10.11 | 10.09 | 97.96 | 98.13 | 99.61 |
| EMNIST Balanced (100% VGG-5 Spinal FC) | homo | 100 | 50 | **85.19** | 85.10 | 84.96 | 84.96 | 2.10 | 2.11 | 84.94 | 84.83 | 89.70 |
| | iid-dq | 100 | 50 | 87.34 | 87.31 | **87.35** | 86.90 | 2.04 | 2.04 | 87.28 | 87.35 | 89.25 |
| | noniid-lds | 100 | 50 | **83.43** | 78.65 | 79.44 | 78.88 | 2.19 | 2.16 | 82.72 | 77.28 | 89.48 |
| | noniid-l18 | 100 | 50 | **85.43** | 81.22 | 81.02 | 80.32 | 2.09 | 2.08 | 82.99 | 82.87 | 89.52 |
| SVHN (100% Resnet-18) | homo | 200 | 80 | **38.90** | 32.69 | 23.08 | 26.30 | 17.12 | 18.64 | 26.12 | 32.34 | 92.75 |
| | iid-dq | 200 | 80 | 65.83 | **66.82** | 64.50 | 51.49 | 17.20 | 18.65 | 58.56 | 65.25 | 93.14 |
| | noniid-lds | 200 | 80 | **32.56** | 28.94 | 28.59 | 30.65 | 10.15 | 10.24 | 30.97 | 32.89 | 92.56 |
| | noniid-l3 | 200 | 80 | **31.45** | 23.92 | 28.62 | 27.47 | 10.21 | 10.71 | 24.14 | 36.66 | 93.24 |
| FEMNIST (100% Resnet-18) | noniid-lds | 3597 | 30 | **54.46** | 44.41 | 51.67 | 46.83 | 21.05 | 21.06 | 55.69 | 50.45 | 89.08 |
| | noniid-lds | 3597 | 100 | **56.36** | 54.61 | 54.04 | 54.07 | 21.05 | 21.06 | 55.69 | 54.21 | 89.08 |
| | noniid-lds | 3597 | 300 | **58.32** | 55.94 | 54.97 | 54.33 | 21.05 | 21.06 | 55.69 | 56.14 | 89.08 |
| CIFAR10 (100% Resnet-50) | homo | 200 | 100 | **32.08** | 32.07 | 30.78 | 30.30 | 10.18 | 9.69 | 32.09 | 32.08 | 88.68 |
| | iid-dq | 200 | 100 | 36.97 | 38.84 | **39.03** | 36.66 | 10.04 | 10.03 | 38.49 | 38.81 | 88.10 |
| | noniid-lds | 200 | 100 | **29.71** | 26.02 | 29.10 | 26.67 | 9.89 | 9.88 | 29.23 | 28.94 | 87.31 |
| | noniid-l4 | 200 | 100 | **34.40** | 32.24 | 30.00 | 30.45 | 10.02 | 9.87 | 33.50 | 34.15 | 89.67 |
| CIFAR100 (100% Resnet-50) | homo | 20 | 12 | **20.84** | 20.58 | 20.65 | 20.48 | 0.99 | 0.99 | 22.84 | 20.85 | 59.81 |
| | iid-dq | 20 | 12 | 47.38 | 47.38 | 47.38 | 25.10 | 1.00 | 0.94 | 47.37 | 47.38 | 60.35 |
| | noniid-lds | 20 | 12 | **16.31** | 15.97 | 16.15 | 15.78 | 0.96 | 0.97 | 18.71 | 15.32 | 60.38 |
| | noniid-l45 | 20 | 12 | **21.29** | 20.56 | 20.26 | 19.97 | 0.92 | 0.91 | 23.68 | 19.61 | 61.74 |

help of selected $K$. Table 1 shows the test performance of selective configurations for different datasets and partition configurations. Here, we choose the VGG-5 Spinal FC model for the EMNIST dataset considering its state-of-the-art performance (except for the extremely large transformer models), the ResNet-50 model for the CIFAR10 and CIFAR100 datasets. Additionally, to demonstrate the effectiveness of our method over heterogeneous models, we simultaneously utilize two types of structures, VGG-5 Spinal FC and Resnet-50, to train the local models for the EMNIST digits and letters datasets.

As we can see, the performance of the *Oracle* method is always the best, since it is the centralized setting that can utilize all parties' data for training the model. Meanwhile, the performance of the *FedAvg* or *MeanAvg* is significantly worse (near random guess), with only test accuracy around 2% for the balanced EMNIST datasets, which validates that directly averaging/fusing well-trained models is not suitable for the one-shot federated learning setting.

As shown in Table 1, for the *homo* partition, the accuracy of different methods are quite similar, which means that they can easily deal with such simple partition case which leads to an iid setting (the same information for all parties) and each client model has similar performance. For the *iid-dq* partition, the *Data Selection (DS)* is the best method for most of the datasets; the single *TOP 1/2* models as in Fig. 4 (b) have the

Table 2: Test Accuracy (%) comparison with state-of-art data-free one-shot federated learning method DENSE for the *EMNIST Digits* Dataset, VGG-5 (Spinal FC) structure when $m$=400 (for `DeDES`, $K = 60$). We compare the test accuracy (%) of the methods for all data partitions.

| Method | DeDES | DENSE | AS | FedAvg |
|--------|-------|-------|-----|--------|
| homo | 98.03 | 92.91 | **98.10** | 10.28 |
| iid-dq | **99.27** | 10.02 | 98.75 | 10.51 |
| noniid-lds | **97.67** | 61.46 | 96.99 | 10.01 |
| noniid-l3 | **98.21** | 55.59 | 97.96 | 10.11 |

Table 3: Complete inspection on ensemble teams for *EMNIST Balanced* dataset with $m = 10, K = 6$, *noniid-lds* partition. Here, we illustrate the rank of the performance of ensemble teams generated by different methods under all $2^{10} = 1024$ teams.

| Method | Rank | Accuracy (%) |
|--------|------|--------------|
| DeDES | **34/1024** | **84.34** |
| AS | 114/1024 | 83.39 |
| CV | 241/1024 | 82.29 |
| DS | 348/1024 | 80.86 |
| RS | 669/1024 | 76.54 |

largest dataset with samples of every class in the label set, which leads to their strong generalization ability under this partition. Therefore, we conclude that the more data we have, the better performance we will get for the iid setting, since the best way is to select $K$ models with top or `top` $K$ largest data sets.

When the data partition is non-iid (*noniid-lds* and *noniid-lk*), we can see that `DeDES` achieves the best performance for most of the datasets, with different $m$ and $K$, which validates the effectiveness of our method. `DeDES` is not as good as the *AS* method for the *CIFAR-100* dataset, this is because *CIFAR-100* has 100 labels, thus the data amount of each individual label for local parties is too tiny to train a generalized model. Under this condition, the *AS* method will get more information than other methods and therefore have better performance. But for other datasets, especially for the EMNIST where all local models are more generalized, `DeDES` will get better performance than others, even the method of all selection.

In summary, our method outperforms other baseline methods for ensemble learning, regardless of the model structures and datasets employed. For the results of heterogeneous models, please refer to the supplementary.

In addition, we compare `DeDES` with the state-of-the-art data-free one-shot federated learning method, DENSE (Zhang et al., 2022). DENSE leverages a generator coupled with the knowledge distillation (KD) technique to train a global model in a data-free way. As shown in Table 2, the performance of `DeDES` in all data partition scenarios is superior to model averaging and DENSE, demonstrating the effectiveness of our method. Furthermore, it can be discerned that while DENSE delivers appreciable results under the homogeneous (*homo*) data distribution, its performance deteriorates to the point of non-convergence in the *iid-dq* scenario, implying that the distribution of training data quantity has a significant impact on DENSE. When dealing with non-iid data partition, the global model's test accuracy attained by DENSE through data generation and knowledge distillation only manages to hit 50%-60% of that secured by ensemble-based methods. This suggests that the data created by the generator is influenced by non-iid data distribution, thereby subsequently impacting the performance of knowledge distillation.

**Complete inspection on ensemble teams.** Given $m = 10$, there are $2^{10} = 1024$ potential ensemble teams available for selection. Table 3 enumerated the accuracy of all 1024 teams and the ranking of selected teams generated by different selection methods. We can see that the ensemble team selected by `DeDES` is ranked higher than other baseline methods, which validates the efficacy of our method.

### 5.4 Impact on Efficiency

As Table 1 shows, `DeDES` is just a little inferior to the ground-truth *All Selection (AS)* method in some cases. However, *AS* is considerably less efficient than our method, and the performance gap between the two methods is small. In other cases, our method even outperforms *AS*. Therefore, it validates that our approach can reduce the inference time significantly for ensemble learning without great sacrifice for the performance.

It is easy to know that when all models are homogeneous, the inference time for ensemble learning (i.e., weighted voting) increases linearly with $K$, i.e., the inference time $T$ for one test sample is $O(K \cdot c)$, where $c$ is a constant inference time for one sample by one model. The inference time for the ensemble team

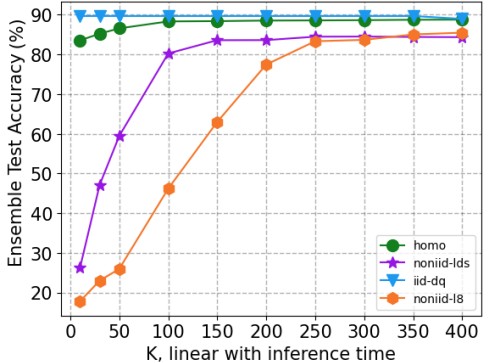

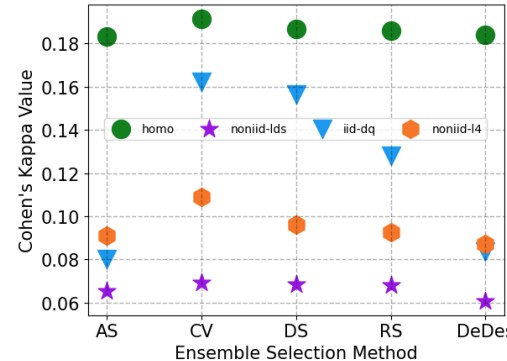

Figure 5: The relationship of $K$ and Ensemble Test Accuracy of `DeDES` for the *EMNIST Letters* Dataset, VGG-5 (Spinal FC) structure when $m$=400. Here we show the accuracy of the AS method when $K = 400$.

Figure 6: The Cohen's Kappa value of ensemble teams selected by different ensemble selection methods for the *CIFAR10* Dataset, Resnet-50 structure when $m$=50, $K = 30$.

generated by the $AS$ method will be $m \cdot c$. Figure 5 shows the relationship between K and ensemble test accuracy of `DeDES`. Note that when $K = m = 400$, we will show the accuracy of the AS method instead of `DeDES` since we typically will not use a selection method such as `DeDES` to select all $K = m = 400$ models.

The experimental results shown in Figure 5 indicate that when $K$ reaches a certain value, the test accuracy will not increase significantly, sometimes even decrease. Therefore, with a suitable $K$ (usually around 50% of $m$), we can substantially reduce our ensemble inference time while achieving good ensemble performance.

**Physical running time comparison and analysis.** To further exemplify the high efficiency of our method, we show the actual physical running time of various selection algorithms in the Appendix, as well as provide an analysis of why and when will our selection method be better than the $AS$ method. To conclude, it is very efficient to use `DeDES` instead of the AS method when the number of test samples exceeds a relatively small threshold.

**Validation for large-scale federated learning system.** To verify the effectiveness and efficiency of our method for large-scale FL system, we tested our method on the FEMNIST dataset which is a benchmark dataset for federated settings that has in total $m = 3597$ clients and is non-i.i.d. Table 1 reveals that when $K = 100 << m$, our method outperforms the "All Selection" method which incorporates all 3597 models into the ensemble team that will yield significant computational load and is highly time-consuming. However, by only choosing 2.78 % (100/3597) of the models by `DeDES`, we can both get better ensemble performance and save approximately 97.22 % (100 - 2.78 %) of the ensemble's execution time. This case substantiates that a model selection method is indispensable for a large-scale federated learning setting.

### 5.5 Clustering/Diversity validation

To validate our clustering results, we compare the Cohen's Kappa (CK) (McHugh, 2012) value of the ensemble teams selected by different methods to measure their diversities. As shown in Figure 6, compared to other baseline methods, the diversity of the ensemble team generated by `DeDES` is higher (lower CK), which also means the agreement of the whole team's models is low. Since we only select one model from every cluster, this finding also indicates that our method can really cluster similar models together, which validates that `DeDES` generates an ensemble team with high diversity. Notably, the $AS$ method also shows high diversity compared to `DeDES` and achieves high ensemble test accuracy. This finding supports the conclusion that greater diversity among models enhances the ensemble's performance.

Table 4: The impact of model filtering method for the *EMNIST Balanced* Dataset, VGG-5 (Spinal FC) structure when $m=200$, $K = 100$ on the noniid-lds partition, where `DeDes_NoMF` refers to our `DeDES` method that does not incorporate model filtering.

| Method | RS | CV | DS | AS | DeDES_NoMF | DeDES |
|---|---|---|---|---|---|---|
| Ensemble Test Accuracy (%) | 74.21 | 74.36 | 78.99 | 79.61 | 80.83 | **81.22** |

Table 5: The impact of model clustering method for the *CIFAR-10* Dataset, Resnet-50 structure when $m=100$, $K = 60$. We compare the test accuracy (%) of the methods for all data partitions.

| Method | DeDES_KMeans | DeDES_Hierarchical | DeDES_Spectral | AS | CV | DS | RS |
|---|---|---|---|---|---|---|---|
| homo | 35.39 | **36.61** | 28.66 | 35.24 | 36.37 | 35.11 | 34.75 |
| iid-dq | 42.86 | 43.82 | 41.51 | 43.79 | 43.85 | **44.40** | 42.13 |
| noniid-lds | **35.76** | 34.36 | 32.86 | 35.04 | 32.91 | 34.12 | 32.72 |
| noniid-l4 | **39.55** | 37.59 | 36.84 | 38.19 | 36.94 | 34.35 | 22.86 |

## 5.6 Ablation Studies

We examine each component's impact on model filtering and clustering methods in this section. Please see the Appendix for additional studies such as model representations, dimension reduction, etc.

### 5.6.1 The Significance of Model Filtering

As illustrated in Table 4, we compare the accuracy of `DeDES` with and without the model filtering algorithm, the results indicate that the use of the model filtering algorithm can effectively eliminate noisy and under-fitting models, leading to an improvement in the final ensemble performance. In contrast to the *AS* method, model filtering can prevent unsuitable models from participating in the ensemble learning process, leading to improved efficiency and efficacy of ensemble learning. Therefore, model filtering is an essential component in conducting ensemble selection for one-shot federated learning.

### 5.6.2 Comparative analysis of clustering methods

Table 5 presents the performance comparison of `DeDES` by implementing various clustering algorithms to form ensemble teams. As we can see, for iid data, hierarchical clustering shows superior performance to other methods, although the discrepancy in accuracy among the different methods remains negligible in an iid setting, as we mentioned in section 5.3. Conversely, under non-iid conditions, KMeans is anticipated to exhibit the highest performance. This expectation is especially true under extreme non-iid circumstances where models trained on similar data are likely to cluster spherically, thus enhancing the effectiveness of the KMeans clustering method over other clustering algorithms.

## 6 Conclusion

In this paper, we propose a novel *data-free diversity-based* framework `DeDES` to address the ensemble selection problem for models under one-shot federated learning. Experimental results show that our method can achieve both better performance and higher efficiency for various model structures and datasets, especially for the non-iid data partitions. To our knowledge, this is the first paper to systematically address the ensemble selection problem under one-shot federated learning setting, which is an essential application for model-centric collaborative machine learning.

## Acknowledgment

This research/project is supported by the National Research Foundation Singapore and DSO National Laboratories under the AI Singapore Programme (AISG Award No: AISG2-RP-2020-018).

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
