# OpenReview forum: "Data-Free Diversity-Based Ensemble Selection for One-Shot Federated Learning"
_TMLR — Accepted by TMLR_

### Review · Reviewer_62Qy · 2023-08-11

**Summary Of Contributions:**

The paper introduces a novel method for one-shot federated learning, where the participants trained the models completely before sending them to the aggregator. Then, the authors propose an ensemble-based method for producing the outputs at inference time. For the ensemble, the authors propose to select a subset of the participants´ models relying on a clustering algorithm that considers the diversity of the models, the number of training points available for each participant and their local performance. The experimental evaluation shows that the proposed approach outperforms other competing methods, including different ensemble-methods, obtaining a more noticeable advantage in scenarios where the datasets

**Audience:**

Yes

**Claims And Evidence:**

Yes

**Requested Changes:**

The method looks interesting and the empirical evaluation, convincing. However, there are certain aspects that can be improved, especially in the description of the algorithm:

• How does the proposed algorithm solve Problem 1? The algorithm is a heuristic that aims to provide a good performance. But there is no clear justification on why the chosen algorithm is a good approximation for solving Problem 1.

• The paragraph about the model representation is too generic. It is fine that different methods can be applied, but the authors should provide more details on concrete techniques that can be more appropriate for this. For example, from the empirical evidence found in the experiments.

• In the description of the algorithm (section 4.1), the authors mentioned a list of different clustering algorithms that can be used. From the results in the experiments, I think that the authors should clarify better or discuss with clustering algorithms are more suitable for the scenario considered. For instance, the observation about the performance of DBSCAN in appendix C.3 is interesting, as well as the results in Section 5.6.2 comparing K-Means and hierarchical clustering.

• In Section 4.1, in the paragraph for representative model selection, the justification for using the score or the number of training data points (depending on the training data balance across clients) is missing and it seems a bit arbitrary. Why is this a good strategy?

• Algorithm 2 not is not well described explained. In case the authors want to keep the main paper short, this part can be moved to the appendix if needed, but, in any case, the details of the algorithm should be explained more clearly.

• The privacy analysis in Section 4.3 requires more depth: the aggregator can perform different types of attacks (e.g., model extraction, membership inference attacks). Same happens if the final model is distributed across the participants of the FL task.

• I did not understand well Section 4.2. Could the authors clarify how is it possible to cluster different structures just by comparison using a DR algorithm? What happens when the different model structures have different number of layers?

• Minor comment: In the related work in Section 2.2 it would be nice to cite the seminal papers on Bagging, boosting, etc. (e.g. Breinman and early works from Schapire in the 90s).


**Strengths And Weaknesses:**

Strengths:

• The authors explore a topic less investigated in the research literature, considering scenarios with one-shot learning for federated learning. In these settings, the challenge is to maximize the performance of the aggregate model by combining the models from the participants appropriately. The topic is interesting, and the application is well justified.

• The method is novel and the way to form the ensemble of models relying on clustering algorithms seems to bring benefits for applications in federated learning with non-IID data. The authors show that using a subset of models using their proposed method is more efficient (both in terms of performance and computational complexity) than using all the participants’ models at test time.

• The experimental evaluation is comprehensive an provide a good overview of the advantages of the method as well as the analysis of its different components. The datasets and models used for the experiments are reasonable and the results are convincing.


Weaknesses:

• Lack of a theoretical analysis or justification on why the proposed algorithm is a good approximation for solving Problem 1.
• Some elements of the proposed method need more justification.

---

> ### Author Response · Authors · 2023-09-10
> **Reply to Reviewer 62Qy (I)**
>
> We appreciate your speedy response and insightful feedback on our paper. We apologize for the delay in our reply, as we had been waiting for comments from all three reviewers based on the policy of TMLR.
>
> Following your suggestions, we have updated the main manuscript as well as the supporting supplementary material. Kindly find below the corresponding links to access the revised documents:
>
> 1. Thr newest version of the main paper: https://openreview.net/pdf?id=ORMlg4g3mG
> 2. The newest version of the supplementary material: https://openreview.net/attachment?id=ORMlg4g3mG&name=supplementary_material
>
> We would also like to provide further clarification on the updates made in response to the "Requested Changes":
>
> **Q1. How does the proposed algorithm solve Problem 1? The algorithm is a heuristic that aims to provide a good performance. But there is no clear justification on why the chosen algorithm is a good approximation for solving Problem 1.**
>
> We added the justification of our algorithm to section B of the supplementary material.
>
> Here we provide a justification for why our method is a good approximation for solving Problem 1.
>
> The following paper presents a series of theorems to measure the relationship between the diversity of an ensemble team and the expected ensemble loss for deep neural networks:
>
> [1] Luis A Ortega, Rafael Cabañas, and Andres Masegosa. Diversity and generalization in neural network
> ensembles. In International Conference on Artificial Intelligence and Statistics, pp. 11720–11743. PMLR, 2022.
>
> As we can see from this paper, Theorem 1 states that the expected loss of an ensemble *decreases* as the diversity measure among model ensembles *increases*. Meanwhile, Theorem 3 in this paper bolsters this evidence, showcasing that the diversity measure proliferates with a decrease in correlation among ensembles.
>
> To sum up, a lower correlation among model ensembles encourages lower expected loss. I.e., an ensemble team with higher diversity will get better ensemble performance.
>
> Back to algorithm 1 in our main paper, we cluster models and subsequently select one from each cluster from Line 5 to Line 13, ensuring that the models in the generated model ensemble are highly uncorrelated among each other thus strengthening the diversity of the ensemble team, and setting the stage for an optimal approximation to Problem 1.
>
> We should note that Theorem 1 and 3 in the above paper are primarily designed for a scenario where data is distributed independently and identically (i.i.d.). However, our experimental results suggest that they perform impressively even in a non-i.i.d data setting, provided that the ensemble group is carefully selected by our method for high diversity. It is a very interesting finding and the relationship between the diversity measure and the expected loss in a non-i.i.d. environment presents a potentially profound area of future theoretical exploration that is worthy of further investigation.
>
> **Q2. The paragraph about the model representation is too generic. It is fine that different methods can be applied, but the authors should provide more details on concrete techniques that can be more appropriate for this. For example, from the empirical evidence found in the experiments.**
>
> We added the discussion of the details about the model representation technique in section A.2 in the supplementary material.
>
> Given a model structure such as Resnet-18, we can choose all or part of the model parameters to represent the model. E.g., as Fig. 2 of the main paper illustrates, we choose to represent models by the parameters of the last layer in every model within the set $\mathcal{M}$. These parameters are preferred as they hold a wealth of individualized information and adequately depict model behavior, particularly for the classifier, and data manifold/space in relation to local training.
>
> If Resnet-18 is used as the model structure for a 10-class dataset, the last layer of its model parameters will be the layer with the name of *linear.weight* and *linear.bias*, which contains 512 $\times$ 10 and 10 parameters, respectively. Then, we will flatten these in total 512 $\times$ 10 + 10 = 51210 parameters into a one-dimensional vector with the size of 51210.
>
> Following this, the model parameter matrix, denoted by $\mathcal{R}\_{\mathcal{M'}}$, is the result of concatenating all these models' one-dimensional vectors and will have the size of $m' \times$ 51210, where $m'$ is the number of models in $\mathcal{M'}$. Prior to leveraging this matrix as the model representation for subsequent clustering, we will employ wide-ranging, commonly used preprocessing mechanisms. These may include MIN-MAX scaling, and/or dimensional reduction on $\mathcal{R}_{\mathcal{M'}}$. This optimizes the representation for better clustering results.

---

> ### Author Response · Authors · 2023-09-10
> **Reply to Reviewer 62Qy (II)**
>
> **Q3. In the description of the algorithm (section 4.1), the authors mentioned a list of different clustering algorithms that can be used. From the results in the experiments, I think that the authors should clarify better or discuss with clustering algorithms are more suitable for the scenario considered. For instance, the observation about the performance of DBSCAN in appendix C.3 is interesting, as well as the results in Section 5.6.2 comparing K-Means and hierarchical clustering.**
>
> We added more discussion about the effect of different clustering algorithms in section E.4 of the supplementary material.
>
> Table 5 of the main paper shows that for iid data, hierarchical clustering outperformed all other methods with regard to performance. Nevertheless, the variance in accuracy between the methods is minuscule in an iid framework, a point we addressed earlier in section 5.3 of the main paper. The rationale behind these results could be attributed to the fact that under iid conditions, the model parameters across clients tend to be similar and positioned closely in terms of the feature space. In such a situation, the data points representing these model parameters might be challenging to separate distinctly. Hence, as KMeans is predominantly proficient in dealing with spherical and well-spaced data, data points in proximity might not exhibit a spherical form or sufficient separation. On the contrary, hierarchical clustering is capable of repeatedly identifying two models that are nearest to each other, regardless of the shape of the data it is working on.
>
> In contrast, under non-iid circumstances, KMeans is expected to outshine other algorithms, particularly under severely non-iid conditions. This behavior is accounted to the fact that, as compared to iid environments, the model parameters are relatively sparse under non-iid settings. Consequently, models trained on similar datasets will likely cluster in a spherical form, enhancing the KMeans clustering method's efficiency over other clustering algorithms.
>
> Moreover, our observations highlighted that Spectral Clustering consistently delivered the lowest performance. This is because Spectral Clustering relies on a presumption that the data encapsulates a low-dimensional structure, which can be exposed via spectral decomposition. In circumstances where the data deviates from this assumption, the performance of Spectral Clustering deteriorates. Hence, for our model selection scenario, Spectral Clustering does not emerge as an ideal choice.
>
> Continuing further in our experimental trajectory, we extensively tested the DBSCAN clustering algorithm. However, DBSCAN failed to segregate the models effectively. This can be attributed to the fact that DBSCAN is fundamentally a density-based clustering method and handles our model parameters, which represent high-dimensional data. As such, the points become considerably sparse, which results in difficulty in defining density.

---

> ### Author Response · Authors · 2023-09-10
> **Reply to Reviewer 62Qy (III)**
>
> **Q4. In Section 4.1, in the paragraph for representative model selection, the justification for using the score or the number of training data points (depending on the training data balance across clients) is missing and it seems a bit arbitrary. Why is this a good strategy?**
>
> We added the discussion about the effect of our representative model selection method in section A.3 of the supplementary material. And here is the reason why we design such a method to select the model from a cluster:
>
> It is widely known that in machine learning and AI, a model's accuracy can often be directly related to the volume of data it has been trained on. In essence, the more diverse and abundant the data available for training, the higher is the potential for the model to comprehend and discern the intricacies of the underlying data. If a client only has a small amount of data, then the trained model might struggle to identify significant patterns or it might overfit to that specific data, meaning it could perform well on that data but poorly on new, unseen data. Inversely, models trained on large volumes of data have a broader foundation to learn from and are typically better at generalizing their findings to new data, thereby often yielding higher accuracy.
>
> We can also validate this conclusion from our experiment results, as depicted in Fig. 4 and Table 1 of the main paper, it is clear that Data Selection (DS) consistently outperforms other methods in most datasets when it comes to the iid-dq partition. This is particularly true when one or a few parties have significantly larger pools of data compared to other clients. The single TOP 1/2 models as in Fig. 4 (b) have the
> the largest dataset with samples of every class in the label set, which leads to their strong generalization ability under this partition. However, when data distribution is more uniform or balanced, referred to as the homogenous partition, the Data Selection method doesn't necessarily offer the best model selection strategy. It's simple to comprehend that in scenarios where data is uniformly distributed among different parties, selecting models purely based on the volume of data becomes ineffective. This is due to the lack of a significant data volume disparity between the client with the highest data volume and the rest. Therefore, a modification to the selection strategy becomes necessary, focusing more on models with the highest validation accuracy rather than just data volume.
>
> We refined a strategy to choose the representative model. If the models within a cluster equivalently distribute the training data, our strategy selects the model with the highest local validation accuracy. Conversely, if the distribution is skewed, the model with the maximum training data is chosen. This strategy was enforced by calculating the ratio of the quantity of local training data for the median model in a cluster to that of the model with the most local training data. If this ratio is less than the prescribed selection threshold $\tau$, the cluster is deemed unbalanced. To substantiate the soundness of our representative model selection technique, we performed an ablation study on three different methodologies:
>
> 1. Mixed: Both data distribution and model validation accuracies were taken into account in this model selection method.
>
> 2. CV: This method consistently chose the model with the highest local validation accuracy.
>
> 3. Data: This method invariably selected the model with the most local data within a cluster.
>
> The data in the table below reveals that the mixed model selection method exceeds other methods across all four data partitions. While the disparity among the selection strategies is not significantly vast (since the representative model selection is just an aspect of the whole model selection algorithm), it still displays superior performance. Not that for the iid-dq data partition, the accuracy of the mixed selection strategy matches that of the data selection strategy. This is a testament to the fact that given a significantly unbalanced dataset, our mixed strategy will invariably choose the model with the most data, mirroring the data selection strategy.
>
>
> | Method | `DeDES_mixed` | `DeDES_CV` | `DeDES_data` |
> | --- | --- | --- | --- |
> | homo | **85.19** | 84.87 | 84.81 |
> | iid-dq | **87.34** | 83.92 | **87.34** |
> | noniid-lds | **83.43** | 83.1 | 83.29 |
> | noniid-l18 | **85.43** | 85.14 | 85.33 |
>
> Table Caption: The impact of the Representative Model Selection method for the EMNIST Balanced Dataset, VGG-5 (Spinal FC) structure when m=100, K = 50, where `DeDES_CV`, `DeDES_data` refers to our DeDES method that always selects the model with the highest score/ model with the most number of local data within a cluster, respectively. `DeDES_mixed` is our mixed selection strategy used in the experiments.

---

> ### Author Response · Authors · 2023-09-10
> **Reply to Reviewer 62Qy (IV)**
>
> **Q5. Algorithm 2 not is not well described explained. In case the authors want to keep the main paper short, this part can be moved to the appendix if needed, but, in any case, the details of the algorithm should be explained more clearly.**
>
> We added more details about how algorithm 2 works in section A.1 of the supplementary material, and we also **provided a figure in section A.1 of the supplementary material to illustrate the algorithm**, we recommend referring to the PDF for a more in-depth understanding.
>
> This technique is essentially an innovative adaptation of the box-plot method, employed specifically for identifying outliers that possess scores beneath the threshold value $\delta$.
>
> Given the range parameters $p_{low}$ and $p_{high}$, in conjunction with a pre-determined scaling factor $s$, the threshold $\delta$ can be computed using the formula $s\times (\hat{\mathcal{S}}\_{m\times p_{high}}-\hat{\mathcal{S}}\_{m\times p_{low}})$.
>
> In the context of our experimental setup, we opted to set $p_{low}$ at 0.25 and $p_{high}$ at 0.75. Under these settings, $\hat{\mathcal{S}}\_{m\times p_{low}}$ and $\hat{\mathcal{S}}\_{m\times p_{high}}$ denote the first quartile (25th percentile) and third quartile (75th percentile), respectively, as typically depicted in a traditional box-plot diagram.
>
> It should be emphasized that the primary aim of this procedure is to eliminate models exhibiting exceptionally low-performance scores, as demonstrated by their respective local validation accuracies in our experiments. As evident from Figure 1 in the supplementary material, any models scoring beneath the threshold $\delta$ are effectively filtered out, thereby excluding them from future clustering pursuits.

---

> ### Author Response · Authors · 2023-09-10
> **Reply to Reviewer 62Qy (V)**
>
> **Q6. The privacy analysis in Section 4.3 requires more depth: the aggregator can perform different types of attacks (e.g., model extraction, membership inference attacks). Same happens if the final model is distributed across the participants of the FL task.**
>
> We provided more discussion about the effect of different clustering algorithms in section A.5 of the supplementary material.
>
> We recognize the potential privacy leakages and are dedicated to mitigating risks in our proposed method. The generator of the GAN is prevented from directly accessing the real data, and our one-shot setting with only a single communication round helps to minimize the exposure of sensitive information. Prior to model submission to the server, we can employ various mechanisms [1, 2] to reinforce model privacy against GAN attacks [3], ensuring enhanced security and data confidentiality.
>
> To thwart model inference attacks, which aim to recover training data from the models, and membership inference attacks that can determine if a particular record was part of the model's training dataset or not, we can choose to incorporate several strategic responses. For instance, the application of differential privacy [4] introduces random noise to the model's outputs, obscuring the model's predictions. Additionally, including the regularization terms [5], such as L1 and L2 regularization, to the model's loss function is also a way to discourage the model from overly fitting the training data, thus safeguarding the training data's privacy.
>
> In terms of model extraction attacks, where the adversary attempts to replicate the targeted machine learning model without direct access to its parameters or training data, a similar approach of differential privacy [6] and regularization [7] could be applied during model training. Furthermore, we can also try to integrate techniques such as watermarking [8], which embeds identifiable information into the model, to detect any unauthorized replication attempts. Other defense policies can also be applied to defend against attack, such as the defense scheme based on the physical unclonable function (PUF) [9], etc.
>
> In summation, by incorporating these diverse measures into our model selection method, we aim to strengthen the protections for model privacy and enhance the security of the model ensemble team.
>
> [1] Aashish Kolluri, Teodora Baluta, and Prateek Saxena. Private hierarchical clustering in federated networks. In Proceedings of the 2021 ACM SIGSAC Conference on Computer and Communications Security, pp. 2342–2360, 2021.
>
> [2] Ruixuan Liu, Yang Cao, Hong Chen, Ruoyang Guo, and Masatoshi Yoshikawa. Flame: Differentially private federated learning in the shuffle model. In Proceedings of the AAAI Conference on Artificial Intelligence, volume 35, pp. 8688–8696, 2021.
>
> [3] Briland Hitaj, Giuseppe Ateniese, and Fernando Perez-Cruz. Deep models under the gan: information leakage from collaborative deep learning. In Proceedings of the 2017 ACM SIGSAC conference on computer and communications security, pp. 603–618, 2017.
>
> [4] Zuobin Xiong, Zhipeng Cai, Daniel Takabi, and Wei Li. Privacy threat and defense for federated learning with non-iid data in aiot. IEEE Transactions on Industrial Informatics, 18(2):1310–1321, 2021.
>
> [5] Solmaz Niknam, Harpreet S Dhillon, and Jeffrey H Reed. Federated learning for wireless communications: Motivation, opportunities, and challenges. IEEE Communications Magazine, 58(6):46–51, 2020.
>
> [6] Haonan Yan, Xiaoguang Li, Hui Li, Jiamin Li, Wenhai Sun, and Fenghua Li. Monitoring-based differential privacy mechanism against query flooding-based model extraction attack. IEEE Transactions on Dependable and Secure Computing, 19(4):2680–2694, 2021.14
>
> [7] Jingtao Li, Adnan Siraj Rakin, Xing Chen, Li Yang, Zhezhi He, Deliang Fan, and Chaitali Chakrabarti. Model extraction attacks on split federated learning. arXiv preprint arXiv:2303.08581, 2023b
>
> [8] Buse GA Tekgul, Yuxi Xia, Samuel Marchal, and N Asokan. Waffle: Watermarking in federated learning. In 2021 40th International Symposium on Reliable Distributed Systems (SRDS), pp. 310–320. IEEE, 2021.
>
> [9] Dawei Li, Di Liu, Ying Guo, Yangkun Ren, Jieyu Su, and Jianwei Liu. Defending against model extraction attacks with physical unclonable function. Information Sciences, 628:196–207, 2023a.

---

> ### Author Response · Authors · 2023-09-10
> **Reply to Reviewer 62Qy (VI)**
>
> **Q7. I did not understand well Section 4.2. Could the authors clarify how is it possible to cluster different structures just by comparison using a DR algorithm? What happens when the different model structures have different number of layers?**
>
> We apologize that this process appears overly complex. In Section A.4 of the supplementary material, we have included a detailed explanation paired with an example to demonstrate the handling of heterogeneous models. Additionally, we have also captured the entire handling process in a comprehensive figure in the supplementary material. Given the limitations of the OpenReview comment box, which restricts the use of complex mathematical equations and cannot show figures, we suggest that you refer to the PDF of the supplementary material for a more profound understanding.
>
> **Q8. Minor comment: In the related work in Section 2.2 it would be nice to cite the seminal papers on Bagging, boosting, etc. (e.g. Breinman and early works from Schapire in the 90s).**
>
> Thank you for your advice, we have added the following related works to the main paper at section 2.2:
>
> [1] Leo Breiman. Bagging predictors. Machine learning, 24:123–140, 1996. doi: 10.1007/BF00058655.
>
> [2] Robert E Schapire. The strength of weak learnability. Machine learning, 5:197–227, 1990. doi: 10.1007/BF00116037.

---

> ### Author Response · Authors · 2023-09-10
> **Friendly Reminder**
>
> Dear Reviewer 62Qy:
>
> We sincerely appreciate the valuable time you have taken to review our paper. We have made considerable efforts to address the concerns raised and have made respective amendments. If there are any parts of our paper that you feel were not adequately explained or remain unclear, we welcome your guidance. Our aim is to ensure that our work is communicated in the most comprehensive and clear manner.
>
> Kindly feel free to outline where additional clarification is needed; we are eager to undertake further revisions as may be necessary.
>
> Best Regards,
>
> Authors.

---

> > ### Comment · Reviewer_62Qy · 2023-10-02
> > **All concerns solved**
> >
> > Thank you very much for the detailed reply to my comments and the effort dedicated to improve the paper. All my concerns have been solved and the new version of the paper looks good.

---

> > > ### Author Response · Authors · 2023-10-02
> > > **Thank you for your precious time in reviewing our paper**
> > >
> > > Dear Reviewer 62Qy,
> > >
> > > We greatly appreciate the time and effort you invested in reviewing our research paper. Thank you!
> > >
> > > Authors.

---

### Review · Reviewer_TB4q · 2023-08-26

**Summary Of Contributions:**

In this paper, the authors proposed a new one-shot federated learning method called DEDES. The proposed method utilizes several tricks, including model filtering, clustering, etc. Empirical performance seems to improve over baselines.

**Audience:**

Yes

**Claims And Evidence:**

No

**Requested Changes:**

- Could the authors provide the physical running time for AS and DeDES for comparison? Explanation on why the proposed method is better than AS is also crucial.
- Could the authors provide ablation studies on different hyperparameter combinations, as well as explanations on how to tune those hyperparameters?

**Strengths And Weaknesses:**

Strengths:

- This paper performs a lot of empirical evaluations.
- The improvement over baseline seems large.


Weaknesses:

- Lack of theoretical guarantees.
- What re the model scores? It does not seem to be defined in the paper.
- Algorithm 1 makes use of 4 tricks. However, it is unclear how much each trick contributes to the utility improvement. While the authors discuss model filtering affects the accuracy in Table 5, what about dimensionality reduction and clustering? And what happens if you always select the model with the highest score / model with the most number of local data within a cluster, rather than using the mixed approach proposed in algorithm 1.
- p_low, p_high and scale s should be the input for algorithm 1 as well
- How does the authors tune the hyperparameters of the proposed method? There are a lot of additional hyperparameters (5) In the proposed method. The authors only seem to discuss how K affects the utility. What about other hyperparameters? More importantly, how do we select these hyperparameters? It is not intuitive to me what is a good grid for searching hyperparameters like \tau, p_low and p_high.
- Table 2 and Figure 5 seems to be showing conflicting results. In Table 2, AS sometimes perform worse compared to DeDES. However, Figure 5 seems to suggest that larger K provides stronger ensemble test accuracy. Could the authors explain this? Also in Figure 5, when m=400, if K could be selected to be 400 as well, isn’t that suggesting that Algorithm 2 filters out no outliers?
- The argument that ‘AS is considerably less efficient’ is not convincing to me. First of all, where are the test data located? From Figure 1, it does not seem that the test data are assumed to live on the edge. In that case, there will be no downward communication from server to client. In that case, while AS takes O(m) inference time, it does not need to spend the huge time complexity spent by DeDES. In particular, it does not need to do clustering, whose time complexity could be really bad when d is large. To me the method only improves upon AS in terms of efficiency when you consider the downward communication cost, which is not explicitly discussed in the paper.

---

> ### Author Response · Authors · 2023-09-10
> **Reply to Reviewer TB4q (I)**
>
> Thank you for your constructive comments on our paper! Following your suggestions, we have updated the main manuscript as well as the supporting supplemental material. Kindly find below the corresponding links to access the revised documents:
>
> - The newest version of the main paper: https://openreview.net/pdf?id=ORMlg4g3mG
> - The newest version of the supplementary material: https://openreview.net/attachment?id=ORMlg4g3mG&name=supplementary_material
>
> We would also like to clarify your questions:
>
> **Q1. Lack of theoretical guarantees.**
>
> While it's true we did not include an explicit theoretical analysis of our algorithm, we have instead presented pertinent justifications that validate our method as an efficient approximation for resolving Problem 1. This proposition is grounded in existing theories, as put forth by notable researchers in our scientific community.
>
> The following paper presents a series of theorems to measure the relationship between the diversity of an ensemble team and the expected ensemble loss for deep neural networks:
>
> [1] Luis A Ortega, Rafael Cabañas, and Andres Masegosa. Diversity and generalization in neural network
> ensembles. In International Conference on Artificial Intelligence and Statistics, pp. 11720–11743. PMLR, 2022.
>
> As we can see from this paper, Theorem 1 states that the expected loss of an ensemble *decreases* as the diversity measure among model ensembles *increases*. Meanwhile, Theorem 3 in this paper bolsters this evidence, showcasing that the diversity measure proliferates with a decrease in correlation among ensembles.
>
> To sum up, a lower correlation among model ensembles encourages lower expected loss. I.e., an ensemble team with higher diversity will get better ensemble performance.
>
> Back to algorithm 1 in our main paper, we cluster models and subsequently select one from each cluster from Line 5 to Line 13, ensuring that the models in the generated model ensemble are highly uncorrelated among each other thus strengthening the diversity of the ensemble team, and setting the stage for an optimal approximation to Problem 1.
>
> We should note that Theorem 1 and 3 in the above paper are primarily designed for a scenario where data is distributed independently and identically (i.i.d.). However, our experimental results suggest that they perform impressively even in a non-i.i.d data setting, provided that the ensemble group is carefully selected by our method for high diversity. It is a very interesting finding and the relationship between the diversity measure and the expected loss in a non-i.i.d. environment presents a potentially profound area of future theoretical exploration that is worthy of further investigation.
>
> **Q2. What are the model scores? It does not seem to be defined in the paper.**
>
> We appreciate you pointing out the clarification required in our statement regarding the usage of the model scores $\mathcal{S}$. In the first paragraph of section 4.1 of the main paper, we did mention that "...each party provides model scores $\mathcal{S}$, which can be their local validation accuracy or prediction confidence." However, we acknowledge the lack of emphasis and positioning of this essential piece of information.
>
> Furthermore, we elaborated on our choice of model score $\mathcal{S}$ within the section "Component Configurations" in the supplementary material. Collecting local validation scores from parties forms the crux of our experiments, and these values are subsequently transmitted to the central model server.
>
> The decision to utilize local validation accuracy as our model score rests on its commonplace use within the machine learning community, paired with its proven efficacy in reflecting model proficiency. Importantly, it does not infringe on the principles of federated learning to relay these scores to the central model server. We hope this sufficiently clarifies our methodology and welcomes any further inquiries.

---

> ### Author Response · Authors · 2023-09-10
> **Reply to Reviewer TB4q (II)**
>
> **Q3.1 Algorithm 1 makes use of 4 tricks. However, it is unclear how much each trick contributes to the utility improvement. While the authors discuss model filtering affects the accuracy in Table 5, what about dimensionality reduction and clustering?**
>
> Actually, except for the discussion of the model filtering method, we did provide ablation studies about clustering and dimensionality reduction methods.
>
> If you are looking at the newest revised version of the PDFs, the ablation study about clustering method comparison can be found in section 5.6.2  in the main paper (Table 5); the ablation study about dimensionality reduction method is in section E.3 of the supplementary material.
>
> Also, here is the link for the revision history of our paper: https://openreview.net/revisions?id=ORMlg4g3mG
>
> If you're checking the old versions, you can find the ablation study about clustering method comparison in section 5.6.2 in the main paper (Table 6); and the ablation study about dimensionality reduction method in section C.2 with Figure 3 of the supplementary material.
>
> I hope the mentioned sections provide thorough clarification.
>
> **Q3.2 And what happens if you always select the model with the highest score/model with the most number of local data within a cluster, rather than using the mixed approach proposed in algorithm 1.**
>
> We appreciate your insight and it's indeed valuable advice to compare the effectiveness of our representative model selection method with other straightforward methods like always selecting the model with the highest score/model with the most local data within a cluster. In response to your suggestion, we have enriched our discussion about the importance of the representative model selection method, and also run additional experiments to check the effect of our representative model selection method, which can be found in section A.3 of the supplementary material.
>
> Moreover, we have copied the results from the supplementary material that provide a comprehensive overview of the effects of different strategies when selecting models within a cluster. The data in the table below reveals that the mixed model selection method exceeds other methods across all four data partitions. While the disparity among the selection strategies is not significantly vast (since the representative model selection is just an aspect of the whole model selection algorithm), it still displays superior performance. Not that for the iid-dq data partition, the accuracy of the mixed selection strategy matches that of the data selection strategy. This is a testament to the fact that given a significantly unbalanced dataset, our mixed strategy will invariably choose the model with the most data, mirroring the data selection strategy.
>
>
> | Method | `DeDES_mixed` | `DeDES_CV` | `DeDES_data` |
> | --- | --- | --- | --- |
> | homo | **85.19** | 84.87 | 84.81 |
> | iid-dq | **87.34** | 83.92 | **87.34** |
> | noniid-lds | **83.43** | 83.1 | 83.29 |
> | noniid-l18 | **85.43** | 85.14 | 85.33 |
>
> Table Caption: The impact of the Representative Model Selection method for the EMNIST Balanced Dataset, VGG-5 (Spinal FC) structure when m=100, K = 50, where `DeDES_CV`, `DeDES_data` refers to our DeDES method that always selects the model with the highest score/ model with the most number of local data within a cluster, respectively. `DeDES_mixed` is our mixed selection strategy used in the experiments.
>
> **Q4. p_low, p_high and scale s should be the input for algorithm 1 as well**
>
> Thank you for your insightful feedback. Based on your suggestions, we have incorporated these input notations into Algorithm 1 in the main paper to provide a clearer and more effective representation of our design. We sincerely appreciate your valuable contribution to improving our work.

---

> ### Author Response · Authors · 2023-09-10
> **Reply to Reviewer TB4q (III)**
>
> **Q5. How does the authors tune the hyperparameters of the proposed method? There are a lot of additional hyperparameters (5) In the proposed method. The authors only seem to discuss how K affects the utility. What about other hyperparameters?  More importantly, how do we select these hyperparameters? It is not intuitive to me what is a good grid for searching hyperparameters like \tau, p_low and p_high.**
>
> The reason why we mainly discuss $K$ is that $K$ is the key to ensemble selection. In practice, it is the final user's decision to determine $K$, depending on their requirements. For example, after evaluating their computational resources, budget and so on. A user may decide to establish an ensemble by selecting 40 models from a pool of 100. In classical ensemble selection literature, this $K$ will be pre-defined by the author.
>
> For other hyperparameters,  like $\tau$ that is used to decide how to select the model within a cluster, and $p_{low}$, $p_{high}$, $s$ are used to decide how many models to be filtered out at the first step of DeDES, we adopted a variant of the widely-used grid search method to search these hyperparameters.  We tested for every combination of specified hyperparameters and the best-performed combination on the validation set will be picked as the final hyperparameters.
>
> The search ranges for different hyperparameters are:
>
> $\tau$: [0.1, 0.2, 0.3, 0.4, 0.5, 0.6, 0.7, 0.8, 0.9], and 0.9 is chosen in our experiments. The logic behind this is as follows: if the model with median local training data in a cluster has less than 90% of the data of the model with most local training data, the cluster is identified as unbalanced. In this case, the model with the most training data is selected. If not, the model with the highest score is chosen.
>
> $p_{low}$: [0.05, 0.10, 0.15, 0.25, 0.30, 0.35]. We finally choose $p_{low}$ to be the 0.25 (25-percentile), just the same as the classical box-plot method since it is widely proven effective.
>
> $p_{high}$: [0.6, 0.65, 0.70, 0.75, 0.80, 0.85]. We finally choose $p_{high}$ to be the 0.75 (75-percentile), just the same as the classical box-plot method.
>
> $s$: [0, 0.5, 1.0, 2.0, 3.0], and 1.0 is ultimately selected. How many models will be filtered out before conducting further clustering will depend on the model score distribution of the clients.
>
> Please refer to section C.3 of the supplementary material to see the whole detail of the hyperparameter tuning method, we also posted the results of the ablation studies on different hyperparameter combinations in section E.1 of the supplementary material to validate the effectiveness of our method.
>
> **Q6.1. Table 2 and Figure 5 seems to be showing conflicting results. In Table 2, AS sometimes perform worse compared to DeDES. However, Figure 5 seems to suggest that larger K provides stronger ensemble test accuracy. Could the authors explain this?**
>
> Thank you for your interesting observations. To begin with, it's important to clarify that in Figure 5, the accuracy of the AS method is still not the best for most of the data partitions, e.g., when $K$=250, the accuracy of our DeDES method is better than the AS for the iid-dq and noniid-lds partition, and with almost the same performance of the homo partition. Here is the reason why initially larger K provides stronger ensemble test accuracy but finally the AS method may not be the best method:
>
> It is generally accepted that the performance of an ensemble improves with the addition of more models when working with a small ensemble size. Increasing the number of models utilizing differing training data increases diversity, thus minimizing the potential for errors across all data points. Each model, with its unique data perspective and learning ability, offers the potential of increased generalization and helps to prevent the propagation of the same error among all models, as each one is likely to make its own distinctive errors. Therefore, this explains the observed performance increase with the increasing value of $K$ in Figure 5, particularly when $K$ is small.
>
> However, this relationship encounters a plateau effect where an increase in $K$ beyond a certain point results in no significant improvement or even deterioration in model performance. Specifically, when the value of $K$ approaches $m$, the total number of models, the diversity of the ensemble tends to stop growing due to the addition of models similar to the existing models in the ensemble team. These new additions tend to make highly correlated mistakes, which limits the number of unique insights. This phenomenon effectively suppresses the benefit of diversification, resulting in reduced ensemble accuracy.
>
> Furthermore, as $K$ increases, the chances of including more low-performing or noisy models also rise. These models, which might exhibit low validation accuracy, do not contribute positively to the ensemble. Instead, they potentially impair the overall performance of the ensemble.

---

> ### Author Response · Authors · 2023-09-10
> **Reply to Reviewer TB4q (IV)**
>
> **Q6.2  Also in Figure 5, when m=400, if K could be selected to be 400 as well, isn’t that suggesting that Algorithm 2 filters out no outliers?**
>
> We appreciate your keen observation and it does call for a crucial point to be clarified. When we referenced Figure 5, the actual values that we present at $K=m=400$ depict the accuracy of the All Selection (AS) method, not the DeDES method. This is because in a typical scenario, users are unlikely to employ a selection method like DeDES to choose all $K=m=400$ models. Hence, in order to find the relationship between the number of selected models and the ensemble test accuracy, we actually represent the accuracy of the All Selection method at $K=m=400$ in Figure 5.
>
> However, this should not be interpreted as if applying $K=400$ to DeDES will not filter any outliers. For a better understanding, we carried out additional experiments setting $K=m=400$ which was consistent with the configuration (EMNIST Letters dataset with VGG-5 models) in Figure 5, and here are the consequential results:
>
> - In the homo data partition, 2 out of 400 models were filtered out by DeDES using Algorithm 2.
>
> - In the iid-dq data partition, 48 out of 400 models were filtered out by DeDES using Algorithm 2.
>
> - In the noniid-lds data partition, 37 out of 400 models were disregarded by DeDES as per Algorithm 2.
>
> - In the noniid-l8 data partition, 46 out of 400 models were removed by DeDES as per Algorithm 2.
>
> We then conducted a comparative performance analysis, juxtaposing the results of the DeDES method with $K=400$, $K=250$, and the AS methods. The comparative results are as follows:
>
> | Partition     | AS          | DeDES (K=400) | DeDES (K=250) |
> |---------------|-------------|---------------|---------------|
> | homo          | 88.54794521 | **88.57534247**   | 88.53082192   |
> | iid-dq        | 88.96387185 | 89.02181322   | **89.35923654**   |
> | noniid-lds    | 84.27537716 | 84.36070722   | **84.38118643**   |
> | noniid-l8     | 85.24051839 | **85.48987467**   | 83.23600724   |
>
> As we can see from the above table, after filtering out noisy models from the model set, DeDES will get better performance than the AS method when $K=400$, despite showing optimal conditions at $K=250$.  This serves to further reinforce our assertion that preemptively filtering out noisy models indeed contributes to the overall improvement in the ensemble performance.
>
> **Q7. The argument that ‘AS is considerably less efficient’ is not convincing to me. First of all, where are the test data located? From Figure 1, it does not seem that the test data are assumed to live on the edge. In that case, there will be no downward communication from server to client. In that case, while AS takes O(m) inference time, it does not need to spend the huge time complexity spent by DeDES. In particular, it does not need to do clustering, whose time complexity could be really bad when d is large. To me the method only improves upon AS in terms of efficiency when you consider the downward communication cost, which is not explicitly discussed in the paper.**
>
> We appreciate your insightful question and here we need to clarify what we mean by 'AS is considerably less efficient'.
>
> > From Figure 1, it does not seem that the test data are assumed to live on the edge. In that case, there will be no downward communication from server to client.
>
> Indeed, the assumption made here is accurate. The test data are not assumed to live on the edge. Thus, there will be no downward communication from server to client. But the communication time is not the focus of our ensemble learning framework, and both our method and AS method will not send models to the edge, thus they all don't have this communication cost.
>
> > While AS takes O(m) inference time, it does not need to spend the huge time complexity spent by DeDES. In particular, it does not need to do clustering, whose time complexity could be really bad when d is large.
>
> Yes, if we only consider the model selection time, the AS method is absolutely quicker than our clustering-based method, especially when $K$ is large.
>
> However, let us not forget that within the context of an ensemble learning system, our target expands beyond merely selecting a fixed ensemble team. We aim to effectively "utilize" this ensemble team in performing inference, carrying out voting procedures, and accomplishing tasks such as image object classification. Thus, a holistic assessment should not just encapsulate the model selection time, but also the subsequent time spent on ensemble learning. That is to say, the total time cost for a method is: $O(selection) + O(ensemble)$. It's important to note that the latter often significantly outweighs the former in terms of time consumption.
>
> (Please see the next comment for a continuation of this answer.)

---

> ### Author Response · Authors · 2023-09-10
> **Reply to Reviewer TB4q (V)**
>
> (Continuing from the previous comment for **Q7**)
>
> **In section A.6 of the supplementary material,** we dive into a more detailed comparison of the time complexity between our method and the AS method. To summarize, the overall time complexity for DeDES and AS will be:
>
> 1. $O(DeDES)$ = $O(selection) + O(ensemble)$ = $O(m \log m + md + m) + O(clustering)+O(Kn_tC)$.
>
> 2. $O(AS)$ = $O(selection) + O(ensemble)$ = $O(1)+O(mn_tC) \approx O(mn_tC)$ .
>
> where $n_t$ is the number of test samples, $C$ is the inference time for a single sample tested by one model.
>
> In order for DeDES to be considered more efficient than AS, the time complexity of AS would need to be greater than that of DeDES. In other words, $O(mn_tC) > O(m \log m + md + m) + O(clustering)+O(Kn_tC)$.
>
> Assuming the clustering method used is KMeans, the inequality can be further refined as: $O(mn_tC) > O(m \log m + md + mKd + m + Kn_tC)$.
>
> I.e., for a chosen $K$, it requires the number of test samples, $n_t$, to be greater than $\frac{m \log m + md + mKd + m}{(m-K)C}$. **This condition is feasibly attainable in practical situations, due to the frequent necessity to test a significant number of samples.** Consequently, $n_t$ can effortlessly transcend a nominal value.
>
> To defend our argument, consider the following example. Table. 1 and Table. 2 display the physical running time comparison for various model selection methods and the inference time for a single test sample on two different datasets, respectively. These were tested on our Linux server, the details of which are provided in section C.4 in the Appendix.
>
> Table 2 shows that processing a test sample through the VGG-5 (Spinal FC) model takes $2.2216 \times 10^{-4}$ second, denoted as $C=2.2216 \times 10^{-4}$. In the case of the EMNIST Letters dataset, we selected $K=200$ from a pool of $m=400$ models for our ensemble. Therefore, gauging by our earlier assertion, DeDES can be considered more efficient than the AS method when: $Time(All\ Selection) + Time(AS\ Ensemble) > Time(DeDES) + Time(DeDES\ Ensemble)$, i.e,
>
> $2.12 \times 10^{-5} + n_t \times 400 \times 2.2216 \times 10^{-4} >  36.73 + n_t \times 200 \times 2.2216 \times 10^{-4}$
>
> Consequently, we get that $n_t >$ 826.66, **suggesting that when we have over 827 test samples at our disposal to carry out ensemble learning, the time efficiency of the DeDES surpasses that of the All Selection method.** This implication becomes particularly encouraging when dealing with the FEMNIST dataset with $K=100$ among $m=3597$ models, whereby $n_t$ merely requires to exceed 116.09 for us to reap both superior performance (performance results can be seen from Table 1. in the main paper) and efficiency through the adoption of the DeDES method in lieu of AS.
>
> Table 1: Physical running time (second) for different model selection methods, KMeans is used as the clustering method.
>
> | Dataset       | Model            | Partition   | m    | K   | DeDES | AS                      |
> | ------------- | ---------------- | ----------- | ---- | --- | ----- | ----------------------- |
> | EMNIST Letters | VGG-5 (Spinal FC) | noniid-l8   | 400  | 200 | 33.73 | 2.12 x 10^-5 |
> | FEMNIST       | Resnet-18        | noniid-lds  | 3597 | 100 | 70.2  | 2.13 x 10^-5 |
>
> Table 2: Physical running time (second) for the inference of one sample.
> | Dataset       | Model            | Partition   | Inference Time          |
> | ------------- | ---------------- | ----------- | ----------------------- |
> | EMNIST Letters | VGG-5 (Spinal FC) | noniid-l8   | 2.2216 x 10^-4 |
> | FEMNIST       | Resnet-18        | noniid-lds  | 1.7291 x 10^-4 |
>
> In practice, we often encounter thousands of test samples in need of inference from the ensemble team. Hence, it is no doubt that applying a model selection method is beneficial for ensemble learning. Table. 3 presents the comprehensive running time for both methods when we have $n_t$ = 100,000 test samples available for voting. The clear observation is that the AS method consumes considerable time due to its model testing and execution of ensemble learning, however, our DeDES provides tangible savings of approximately 49.59\% of the time than AS for the EMNIST Letters dataset, and a remarkable **97.1\%** of the time for the FEMNIST dataset than AS with no compromise on performance. This empirically attests the superior efficiency and effectiveness of our proposed method.
>
> Table 3: Physical running time (second) comparison of DeDES and AS for the whole ensemble selection process, including model selection and inference of ensemble learning where KMeans is used as the clustering method for DeDES on $n_t$=100,000 test samples.
>
> | Dataset       | Model            | Partition  | m    | K   | DeDES  | AS        |
> | ------------- | ---------------- | ---------- | ---- | --- | ------ | --------- |
> | EMNIST Letters | VGG-5 (Spinal FC) | noniid-l8  | 400  | 200 | 4479.93 | 8886.40  |
> | FEMNIST       | Resnet-18        | noniid-lds | 3597 | 100 | 1799.3 | 62195.73  |

---

> ### Author Response · Authors · 2023-09-10
> **Reply to Reviewer TB4q (VI)**
>
> Now we consider your **Requested Changes**:
>
> **1. Could the authors provide the physical running time for AS and DeDES for comparison? Explanation on why the proposed method is better than AS is also crucial.**
>
> In the answer of **Q7**, we have offered a thorough examination of the physical running times for AS and DeDES, accounting for both their selection time and inference time, as well as the comprehensive ensemble learning procedure on $n_t=100,000$ test data points.
> The results are incorporated into Section A.6 of the supplementary material, specifically within Tables 1 through 3. Additionally, the comprehensive justification for why the proposed method is better than AS is also documented in Section A.6 of the supplementary material.
>
> **2. Could the authors provide ablation studies on different hyperparameter combinations, as well as explanations on how to tune those hyperparameters?**
>
> As mentioned in the answer to **Q5**, we already provided brief explanations on how to tune hyperparameters of DeDES. More details are documented in section C.3 of the supplementary material.
>
> For the ablation studies on different hyperparameter combinations, we posted the results here, and also in section E.1 of the supplementary material. Please refer to the PDF of the supplementary material to see the whole findings and analysis.
>
> Table 4 presents the comparative analysis of seven distinct hyperparameter combinations. Initially, one can observe that the fluctuation in hyperparameters does not bring a substantial change in performance, thereby solidifying the stability of our DeDES approach. Simultaneously, Table 4 reaffirms that regardless of the hyperparameters selected, the homo data partition will consistently deliver identical performance. This consistency can be attributed to the homo data partition generating models with nearly the same model parameters, thereby ensuring that no matter how many models we filter out or what model will we choose within a cluster, the eventual ensemble performance remains largely similar.
>
> Observing other data partitions reveals a similar influence of varying hyperparameters. For instance, the necessity for a model filtering algorithm becomes apparent when comparing combinations 1, 2 and 3, with $s$ = 0 implying that no models are filtered out, resulting in a performance decrement. When comparing combinations 1, 4, and 5, it is implied that there's a requirement for an effective threshold $\tau$ to determine the degree of data imbalance among all clients. A $\tau$ value of 0.9 signifies data-sensitivity and a tendency to select a model with a significant amount of local training data. Nevertheless, it is not advisable to set $\tau$=1, as it implies always select the model with the highest quantity of data within a cluster, which could potentially overshadow models with superior model scores, leading to a dip in performance, echoing the findings in section A.3 of the supplementary material.
>
> An examination of combinations 1, 6, and 7 indicates that the value of the threshold pair ($p_{low}, p_{high}$) doesn't significantly impact performance, with the commonly utilized 25-percentile and 75-percentile exhibiting the best performance among all combinations.
>
>
>
> Table 4: Performance comparison of choosing different hyperparameter combinations for the SVHN dataset, Resnet-18 structure when $m=200$, $K=80$. Here we provide 7 hyperparameter combinations to show the effects of these hyperparameters.
>
> | ID | $\tau$ | $p_{low}$ | $p_{high}$ | $s$  | homo | iid-dq | noniid-lds | noniid-l3 |
> |----|-------|-----------|------------|-----|------|--------|-------------|-----------|
> | 1  | 0.9   | 0.25      | 0.75       | 1   | **38.90** | **65.83**  | **32.56**      | **31.45**  |
> | 2  | 0.9   | 0.25      | 0.75       | 0   | **38.90** | 65.46  | 32.21      | 30.92   |
> | 3  | 0.9   | 0.25      | 0.75       | 2   | **38.90** | 65.82  | 32.50      | 31.45   |
> | 4  | 0.1   | 0.25      | 0.75       | 1   | **38.90** | 65.03  | 32.41      | 31.07   |
> | 5  | 0.5   | 0.25      | 0.75       | 1   | **38.90** | 65.35  | 32.51      | 31.11   |
> | 6  | 0.9   | 0.35      | 0.85       | 1   | **38.90** | 65.82  | 32.54      | **31.45**  |
> | 7  | 0.9   | 0.05      | 0.6        | 1   | **38.90** | 65.81  | 32.53      | 31.44   |

---

> ### Author Response · Authors · 2023-09-10
> **Friendly Reminder**
>
> Dear Reviewer TB4q:
>
> We sincerely appreciate the valuable time you have taken to review our paper. We have made considerable efforts to address the concerns raised and have made respective amendments. If there are any parts of our paper that you feel were not adequately explained or remain unclear, we welcome your guidance. Our aim is to ensure that our work is communicated in the most comprehensive and clear manner.
>
> Kindly feel free to outline where additional clarification is needed; we are eager to undertake further revisions as may be necessary.
>
> Best Regards,
>
> Authors.

---

### Review · Reviewer_VK3t · 2023-09-05

**Summary Of Contributions:**

The paper proposes a novel data-free diversity-based framework, DeDES, to address the ensemble selection problem with diversity consideration for models under the one-shot federated learning setting. The authors demonstrate that their method can achieve better performance and higher efficiency over various datasets and model structures. The proposed framework is evaluated on several benchmark datasets, and the experimental results show that it outperforms existing state-of-the-art methods in terms of accuracy and efficiency. The paper is well-structured and clearly presents the proposed framework and its experimental results. The authors provide a comprehensive literature review and a detailed explanation of the proposed method, which makes it easy to understand and replicate. The experimental results are presented in a clear and concise manner, and the authors provide a thorough analysis of the results.
Overall, the paper presents a novel and effective approach to the ensemble selection problem in federated learning. The proposed framework has the potential to be applied to a wide range of applications and can significantly improve the performance and efficiency of federated learning systems.

**Audience:**

Yes

**Claims And Evidence:**

Yes

**Requested Changes:**

1). Include more datasets such as FEMNIST.
2). More thorough related work discussions are suggested to include the most recent publications
3). Include the computational complexity and resource requirements (communication costs) of the proposed framework and experiments on it.

**Strengths And Weaknesses:**

Pros:
- The paper proposes a novel data-free diversity-based framework, DeDES, to address the ensemble selection problem with diversity consideration for models under the one-shot federated learning setting.
- The authors provide a comprehensive literature review and a detailed explanation of the proposed method, which makes it easy to understand and replicate.
- The experimental results show that the proposed framework outperforms existing state-of-the-art methods in terms of accuracy and efficiency.
- The proposed framework is evaluated on several benchmark datasets, which demonstrates its effectiveness and potential for real-world applications.
- The paper is well-structured and clearly presents the proposed framework and its experimental results.
Cons:
- The experimental results are limited to a specific set of benchmark datasets and may not generalize to other datasets or real-world applications, such as FEMNIST.
- More thorough related work discussions are suggested to include most recent publications, such as:
Personalized Federated Learning under Mixture of Distributions.
International Conference on Machine Learning (ICML'23), 2023.
- The paper does not provide a detailed discussion of the computational complexity and resource requirements of the proposed framework, which may limit its scalability in large-scale federated learning systems.

---

> ### Author Response · Authors · 2023-09-10
> **Reply to Reviewer VK3t (I)**
>
> Thank you for your valuable advice on our paper! Following your suggestions, we have updated the main paper as well as the supporting supplemental material. Kindly find below the corresponding links to access the revised documents:
>
> - The newest version of the main paper: https://openreview.net/pdf?id=ORMlg4g3mG
> - The newest version of the supplementary material: https://openreview.net/attachment?id=ORMlg4g3mG&name=supplementary_material
>
> We would like to impart additional clarity concerning the revisions that we've conducted following your suggested changes. As your request changes directly pertain to the perceived weaknesses in our paper, we have taken the liberty to respond to them collectively:
>
> **1. The experimental results are limited to a specific set of benchmark datasets and may not generalize to other datasets or real-world applications, such as FEMNIST.**
>
> **Requested Change: Include more datasets such as FEMNIST.**
>
> We totally agree that including more large-scale, real-world datasets is valuable to prove the good generalization capability of our method. While we acknowledge the need for comprehensive experimental results, we only have five days to run experiments since the system requires us the submit the revised paper by Monday, thus we don't have too much time to run more datasets. Regardless, we have managed to run experiments on two additional datasets:
>
> 1. FEMNIST: a benchmark dataset for federated learning that contains in total 3597 clients, whose data distribution is non-i.i.d among these clients.
> 2. SVHN: The Street View House Numbers (SVHN) dataset is a public dataset that contains real-world, full-color images of house numbers extracted from Google Street View images. We use this dataset to validate that our method is generalizable and can be applied to real-world applications.
>
> We partition the SVHN dataset into 200 parties based on the four types of data partitions mentioned in the main paper. Then, we train the models with the Resnet-18 structure for both datasets. Here are the corresponding results, **which are also added to Table 1 of the main paper**:
>
> | Dataset | Partition | m | K | DeDES | CV | DS | RS | FedAvg | MeanAvg | AS | LD | Oracle |
> | --- | --- | --- | --- | --- | --- | --- | --- | --- | --- | --- | --- | --- |
> | FEMNIST | noniid-lds | 3597 | 30 | **54.46** | 44.41 | _51.67_ | 46.83 | 21.05 | 21.06 | 55.69 | 50.45 | 89.08 |
> | FEMNIST | noniid-lds | 3597 | 100 | **56.36** | _54.61_ | 54.04 | 54.07 | 21.05 | 21.06 | 55.69 | 54.21 | 89.08 |
> | FEMNIST | noniid-lds | 3597 | 300 | **58.32** | _55.94_ | 54.97 | 54.33 | 21.05 | 21.06 | 55.69 | 56.14 | 89.08 |
> | SVHN | homo | 200 | 80 | **38.90** | _32.69_ | 23.08 | 26.30 | 17.12 | 18.64 | 26.12 | 32.34 | 92.75 |
> | SVHN | iid-dq | 200 | 80 | _65.83_ | **66.82** | 64.50 | 51.49 | 17.20 | 18.65 | 58.56 | 65.25 | 93.14 |
> | SVHN | noniid-lds | 200 | 80 | **32.56** | 28.94 | 28.59 | _30.65_ | 10.15 | 10.24 | 30.97 | 32.89 | 92.56 |
> | SVHN | noniid-l3 | 200 | 80 | **31.45** | 23.92 | _28.62_ | 27.47 | 10.21 | 10.71 | 24.14 | 36.66 | 93.24 |
>
> Basically, the conclusion remains unchanged from other datasets, but we are surprised to see that for the FEMNIST dataset, when $K=100 << m $ that is 3597, our method outperforms the "All Selection" method which incorporates all 3597 models into the ensemble team that will yield significant computational load and is highly time-consuming. However, by only choosing 2.78\% (100/3597) of the models by DeDES, we can both get better ensemble performance and save approximately 97.22\% (100 - 2.78\%) of the ensemble's execution time. This case substantiates that a model selection method is indispensable for a large-scale federated learning setting.

---

> ### Author Response · Authors · 2023-09-10
> **Reply to Reviewer VK3t (II)**
>
> **2. More thorough related work discussions are suggested to include most recent publications, such as: Personalized Federated Learning under Mixture of Distributions. International Conference on Machine Learning (ICML'23), 2023.**
>
> **Requested Change: More thorough related work discussions are suggested to include the most recent publications**
>
> Indeed, we recognize the importance of including up-to-date literature in our research. In response to this, we added the following 8 works of literature to Section 2 of our main paper with some additional discussion:
>
> - Wu, Yue, et al. "Personalized Federated Learning under Mixture of Distributions." International Conference on Machine Learning (ICML'23), 2023.
>
> - Jhunjhunwala, Divyansh, Shiqiang Wang, and Gauri Joshi. "Towards a Theoretical and Practical Understanding of One-Shot Federated Learning with Fisher Information." Federated Learning and Analytics in Practice: Algorithms, Systems, Applications, and Opportunities. 2023.
>
> - Feng, Chun-Mei, et al. "Learning Federated Visual Prompt in Null Space for MRI Reconstruction." Proceedings of the IEEE/CVF Conference on Computer Vision and Pattern Recognition. 2023.
>
> - Heinbaugh, Clare Elizabeth, Emilio Luz-Ricca, and Huajie Shao. "Data-Free One-Shot Federated Learning Under Very High Statistical Heterogeneity." The Eleventh International Conference on Learning Representations. 2022.
>
> - Diao, Yiqun, Qinbin Li, and Bingsheng He. "Towards Addressing Label Skews in One-Shot Federated Learning." The Eleventh International Conference on Learning Representations. 2022.
>
> - Gong, Xuan, et al. "Ensemble attention distillation for privacy-preserving federated learning." Proceedings of the IEEE/CVF International Conference on Computer Vision. 2021.
>
> - Lin, Tao, et al. "Ensemble distillation for robust model fusion in federated learning." Advances in Neural Information Processing Systems 33 (2020): 2351-2363.
>
> - AbdulRahman, Sawsan, et al. "FedMCCS: Multicriteria client selection model for optimal IoT federated learning." IEEE Internet of Things Journal 8.6 (2020): 4723-4735.
>
>
> **3. The paper does not provide a detailed discussion of the computational complexity and resource requirements of the proposed framework, which may limit its scalability in large-scale federated learning systems.**
>
> **Requested Change: Include the computational complexity and resource requirements (communication costs) of the proposed framework and experiments on it.**
>
> In Section 4.3 of our main manuscript, we've offered detailed insights into the Complexity Analysis and Resource Requirements. Supplementary material further investigates Computational/Time Complexity in Section A.6, providing a more extensive discussion for your consideration. Additionally, we've demonstrated our method's physical running time in Section A.6 of these supplementary materials.
>
> To clarify, since our approach adheres to the one-shot federated learning scheme, which inherently requires every client to communicate with the server **only once** - when they transmit their respective models. Consequently, the sole communication cost incurred by every client is the transmission of its meticulously trained model to the central server. To illustrate, a client, such as a smartphone, may train a Resnet model with a 44-MB size, and dispatch it to our server via the internet. This task is not a challenge for modern devices. We would like to emphasize that our usual protocol does not impose any fixed constraints on the computational resources designated for the central server. This is based on our fundamental belief that the central server possesses an exceptional computational capability. Hence, in discussing communication costs and computational resources, our method exhibits flexibility and scalability. It can be comfortably expanded to accommodate a large-scale federated learning system without compromising efficiency.
>
> In terms of time complexity, we conclude that the overall time complexity of DeDES is: $O(DeDES)$ = $O(selection) + O(ensemble)$ = $O(m \log m + md + m) + O(clustering)+O(Kn_tC)$, where $n_t$ is the number of test samples, $C$ is the inference time for a single sample tested by one model.  In practice, selecting $K=100$ models from a pool of $m=3597$ Resnet-18 models becomes significantly efficient in terms of both time and memory utilization. It merely takes approximately 75 seconds with a refined consumption of 2GB of RAM without GPU utilization when executed on a 256-core AMD EPYC 7742 64-Core Processor clocking at a speed of 3.4GHz.

---

> ### Author Response · Authors · 2023-09-10
> **Friendly Reminder**
>
> Dear Reviewer VK3t:
>
> We sincerely appreciate the valuable time you have taken to review our paper. We have made considerable efforts to address the concerns raised and have made respective amendments. If there are any parts of our paper that you feel were not adequately explained or remain unclear, we welcome your guidance. Our aim is to ensure that our work is communicated in the most comprehensive and clear manner.
>
> Kindly feel free to outline where additional clarification is needed; we are eager to undertake further revisions as may be necessary.
>
> Best Regards,
>
> Authors.

---

### Decision · Action_Editor_iqup · 2023-11-06

**Recommendation:** Accept as is

**Comment:**

The main contribution of this paper is to propose an empirical approach to the ensemble selection problem under one-shot federated learning. The paper performed a comprehensive evaluation of the proposed methods over multiple standard benchmarking tasks and various aspects (e.g., computational overheads).

Two reviewers are positive about the contribution(s), and one reviewer had more concerns than strengths. However, the authors have done a good job of addressing the question and concerns, and therefore, the AE would like to recommend “Accept.”

**To the authors:** The AE encourages the authors to incorporate the promised revisions that have not been incorporated yet in the final version of the paper.

**Audience:**

I believe the audience of this paper could be limited, e.g., to those interested in seeking an ensemble approach and improving its performance. But as finding the optimal ensemble is purely empirical, there's no guarantee that the audience will see the same improvement in the domain(s) or application(s) of the audience's interests.

**Claims And Evidence:**

The claims of this paper lack theoretical guarantees, but the paper provides a comprehensive empirical evaluation to back the claims.

---

> ### Author Response · Authors · 2023-11-15
> **Camera Ready Version and Revision**
>
> We have submitted the revised camera-ready version according to the TMLR format.
>
> Furthermore, we have meticulously revised our paper to incorporate all promised revisions. We are grateful to our reviewers and the action editor for their invaluable and insightful suggestions.